# The origin of Asian Monsoons: a modelling perspective

Delphine Tardif[1], Frédéric Fluteau[1], Yannick Donnadieu[2], Guillaume Le Hir[1], Jean-Baptiste Ladant[3], Pierre Sepulchre[4], Alexis Licht[5], Fernando Poblete[6], Guillaume Dupont-Nivet[7,8]

[1] Université de Paris, Institut de physique du globe de Paris, CNRS, 75005 Paris, France

[2] Aix-Marseille Univ, CNRS, IRD, Coll France, INRA, CEREGE, Aix-en-Provence, France.

[3] University of Michigan, Ann Arbor, MI, USA

[4] Laboratoire des Sciences du Climat et de l'Environnement, LSCE/IPSL, CEA-CNRS-UVSQ, Université Paris-Saclay, 91191 Gif-sur-Yvette, France

[5] University of Washington, Seattle, USA

[6] Departamento de Geología, Universidad de Chile, Santiago, Chile

[7] Univ. Rennes, CNRS, Géosciences Rennes, 35000 Rennes, France

[8] Institute of Geosciences, Universität Potsdam, Germany

**Abstract**. The Cenozoic inception and development of the Asian monsoons remain unclear and have generated much debate, as several hypotheses regarding circulation patterns at work in Asia during the Eocene have been proposed in the last decades. These include a) the existence of modern-like monsoons since the early Eocene; b) that of a weak South Asian Monsoon (SAM) and little to no East Asian Monsoon (EAM) or c) a prevalence of the Inter Tropical Convergence Zone (ITCZ) migrations, also referred to as Indonesian-Australian Monsoon (I-AM). As SAM and EAM are supposed to have been triggered or enhanced primarily by Asian paleogeographic changes, their possible inception in the very dynamic Eocene paleogeographic context remains an open question, both in the modeling and field-based communities. We investigate here Eocene Asian climate conditions using the IPSL-CM5A2 earth system model and revised paleogeographies. Our Eocene climate simulation yields atmospheric circulation patterns in Asia substantially different from modern. A large high-pressure area is simulated over the Tethys ocean, which generates intense low tropospheric winds blowing southward along the western flank of the proto Himalayan Tibetan plateau (HTP) system. This low-level wind system blocks, to latitudes lower than 10°N, the migration of humid and warm air masses coming from the Indian Ocean. This strongly contrasts with the modern SAM, during which equatorial air masses reach a latitude of 20-25°N over India and southeastern China. Another specific feature of our Eocene simulation is the widespread subsidence taking place over northern India in the mid troposphere (around 5000 m), preventing deep convective updraft that would transport water vapor up to the condensation level. Both processes lead to the onset of a broad arid region located over northern India and over the HTP. More humid regions of high seasonality in precipitations encircle this arid area, due to the prevalence of the Inter Tropical Convergence Zone (ITCZ) migrations (or Indonesian-Australian Monsoon, I-AM) rather than monsoons. Although the existence of this central arid region may partly result from the specifics of our simulation (model dependence, paleogeographic uncertainties) and has yet to be confirmed by proxy records, most of the observational evidence for Eocene monsoons are located in the highly seasonal transition zone between the arid area and the more humid surroundings. We thus suggest that a zonal arid climate prevailed over Asia before the initiation of Monsoons that most likely occurred following Eocene paleogeographic changes. Our results also show that precipitation

seasonality should be used with caution to infer the presence of a monsoonal circulation and that the collection of new
data in this arid area is of paramount importance to allow the debate to move forward.
**1. Introduction**
Monsoons are characterized by highly seasonal precipitations, with a dry season in winter and a wet season in summer,
along with a seasonal wind inversion (Wang and Ding, 2008). From this definition, several broad monsoonal regions can
be identified over the globe (Zhang and Wang, 2008; Zhisheng et al., 2015). A prominent member is the Asian Monsoon
system, which covers several smaller monsoonal regions (Wang and LinHo, 2002). The South Asian Monsoon (SAM) is
characterized by dry winters and wet summers with rainfall occurring from May (in southern India and Southeastern Asia)
to July (in northwestern India). The East Asian Monsoon (EAM) presents more contrasted seasons with cold and dry
winters due to the presence of the Siberian High, and hot and wet summers with rainfall maxima from May (southeastern
China) to July (northeastern China). The Indonesian-Australian Monsoon (I-AM), mirrored in the North by the mostly
oceanic Western Northern Pacific Monsoon (WNPM), results from the seasonal migration of the Inter Tropical
Convergence Zone (ITCZ) and generates rainfall from April to August (over southeastern Asia and western Pacific) and
from November to February (over Indonesia and northern Australia).
The ITCZ is an intrinsic characteristic of the Earth's climate and the WNPM and I-AM have therefore probably occurred
throughout Earth's history (Spicer et al., 2017). On the other hand, the triggering factors of both SAM and EAM are more
complex and remain debated. Although the SAM is also related to the migration of the ITCZ, it is thought to be enhanced
by orographic insulation provided by the Himalayas (Boos and Kuang, 2010), by the overheating of the Tibetan Plateau
(TP) in summer, and by the generation of a strong Somali jet (Molnar et al., 2010), which might itself be amplified by the
East African coast's orography (Bannon, 1979), although this view has been challenged (Wei and Bordoni, 2016).
Another characteristic feature of the SAM is a strong shear zone between the 850 and 200 mb zonal winds (Webster and
Yang, 1992). In contrast, the EAM is an extra-tropical phenomenon, where winter and summer monsoons are mainly
triggered by differential cooling and heating between the huge Asian continental landmass and the western Pacific Ocean,
even though it has been suggested that the EAM might also be affected by the Somali Jet strength and TP uplift (Tada et
al., 2016).
The inception of the SAM and EAM has been proposed to have occurred during the early Miocene (Guo et al., 2002) or
the latest Oligocene (Sun and Wang, 2005) but recent field observations have suggested an earlier inception, as soon as
the middle to late Eocene (~40 Ma). These studies rely on different indices such as a) records of high seasonality in
precipitations from paleovegetation and sedimentary deposits in China (Quan et al., 2012; Sorrel et al., 2017; Q. Wang et
al., 2013) and Myanmar (Licht et al., 2015); b) $\delta^{18}O$ measurements showing high annual variability in water availability
in oyster shells from the Tarim Basin (Bougeois et al., 2018; Ma et al., 2019), in mammals tooth enamel and gastropod
shells from Myanmar (Licht et al., 2014). These findings postpone the initiation of the Asian monsoons by about 20 Myr
and, given the strong dependence of both SAM and EAM to paleogeography, orography and temperature gradients, raise
a challenge of understanding the triggering factors of these complex atmospheric systems in the climatic and
paleogeographic context of the middle to late Eocene.
Indeed, the second half of the Eocene, referred to as "doubthouse", is a key period in the transition from the warm ice-
free early Eocene greenhouse to colder icehouse initiated in the early Oligocene (Liu et al., 2009). It witnessed profound
climatic modifications, such as a global cooling and drying, the possible onset of the Antarctic Circumpolar Current

(ACC) and a large-scale glaciation in Antarctica (Sijp et al., 2014), hence prefiguring the dawning of modern climatic features. Moreover, important paleogeographic changes took place in the Late Eocene in Asia following the collision between the Eurasian and Indian continents, that might have significantly impacted both regional and global climate; including a) two Paratethys sea retreat with fluctuations phases between 46 and 36 Ma (Meijer et al., 2019); b) the drying and subsequent closure of the India foreland basin (Najman et al., 2008) and c), continued uplift of the Tibetan Plateau (Kapp and DeCelles, 2019).

If no consensus has been reached so far regarding the possibility of modern-like SAM and EAM in the Eocene, and on the mechanisms at stake during this period, several conjectures have emerged in the last decades. With the NCAR CCSM3 fully coupled model, Huber and Goldner, (2012) suggest that the global monsoon system (including the Asian monsoons) prevailed throughout the Eocene. Using a Late Eocene configuration and the Fast Ocean Atmosphere Model (FOAM) along with LMDZ atmosphere model, Licht et al., (2014) postulate the existence of the Asian monsoons in the late Eocene and show that orbital forcing might even trigger monsoons stronger than the modern ones. Other studies have also inferred the existence of the Asian monsoons in the late Eocene on the basis of sensitivity experiments deriving mainly from modern geographic configurations (Roe et al., 2016; Zoura et al., 2019).

Other studies, although more focused on the EAM, are more cautious regarding the prevalence of the monsoons in the Eocene. Zhang et al., (2012) using FOAM suggest that early Eocene Asia was dominated by steppe/desert climates, with a stable SAM but only an intermittent EAM depending on the orbital forcing. Li et al., (2018) and Zhang et al., (2018) perform late Eocene climate simulations with the low-resolution NorESM-L Earth System Model (ESM) and the NCAR CAM4 atmospheric model and further show that the wind and precipitation patterns simulated in eastern China are not comparable to the modern EAM.

A third theory has also recently been suggested based on both modeling work (Farnsworth et al., 2019) and leaf physiognomic signatures from vegetation deposits from southeastern China, which is a region nowadays experiencing a mixed influence of EAM, I-AM and SAM (Herman et al., 2017; Spicer et al., 2016). They show that the fossil floras from the Maoming and Changchang basins display more similarities with modern floras submitted to the influence of I-AM than to that of any other monsoon, hence suggesting that ITCZ migration could have been the main driver of precipitation seasonality in the late Eocene.

The discrepancies between these different conjectures are hardly straightforward, given the variety of modeling framework, model resolution and boundary conditions involved in the aforementioned studies, let alone considering the possible biases of any model. From an observational perspective, available paleoclimatic markers in Asia are also divided between proxies suggesting the presence of Eocene monsoons and others that do not. However, the uncertainties associated with the climatic controls of the diverse proxies used to infer the existence of Eocene Asian monsoons often hamper the unequivocal assignment of the proxy signals to the monsoons.

In this study, we first test the robustness of our ESM by analyzing monsoonal circulations for modern conditions. The use of an ESM here is particularly indicated given the importance of atmosphere-SST interactions in monsoon circulation. We then simulate the late-middle Eocene (42 to 38 Ma) climate using a 40 Ma paleogeographic reconstruction. First, we perform a global model-data comparison with both continental and marine temperatures, allowing us to demonstrate the ability of our model to simulate the late Eocene climate at the first order. Second, we analyze atmospheric circulation patterns over Asia and highlight potential (di-)similarities with modern circulation. We finally focus on the atmospheric dynamics and on the hydrological cycle features occurring over the Asian continent during the late Eocene, and discuss the possible reasons behind the discrepancies observed between the different existing hypotheses.

## 2. Model and methods

### 2.1. Model description and validation

IPSL-CM5A2 (Sepulchre et al., 2019) is composed of the atmospheric LMDZ5 model (Hourdin et al., 2013), the land surface ORCHIDEE model (Krinner et al., 2005) and the NEMO model including oceanic, biogeochemical and sea-ice components (Madec, 2016). The atmospheric grid has a resolution of 3.75° (longitude) by 1.89° (latitude) and 39 vertical layers from the surface up to 40 km high and the tripolar oceanic grid has a resolution varying between 0.5° to 2° and 30 vertical layers. The continuity of the processes at the interface between ocean and atmosphere is ensured by the OASIS coupler (Valcke et al., 2006). The land surface ORCHIDEE model is coupled to the atmosphere with a 30 mn time-step and includes a river runoff module to route the water to the ocean (d'Orgeval et al., 2008). Vegetation is simulated through eleven Plant Functional Types (hereafter PFT): eight forest PFTs, one bare soil PFT and two grasses PFTs, one coding for $C_3$ grasses and the other one for $C_4$ grasses (Poulter et al., 2011). As the $C_4$ plants are known to expand during the late Miocene (Cerling et al., 1993), this last PFT was deactivated.

IPSL-CM5A2 is an updated version of IPSL-CM5A (Dufresne et al., 2013), which was already used in paleoclimate studies for the Quaternary (Kageyama et al., 2013) and the Pliocene (Contoux et al., 2012; Tan et al., 2017). It relies on more recent versions of each component, and has been re-tuned to reduce the IPSL-CM5A global cold bias. Apart from retuning - that is based on a new auto conversion threshold for water in cloud - and various improvements in energy conservation, the LMDZ component is of IPSL-CM5A2 has the same physics and parameterizations than IPSL-CM5A. Jet position and AMOC have been improved, together with the sea-ice cover. IPSL-CM5A2 also benefits from higher parallelization (namely MPI-OpenMP in the atmosphere), which improves the model scalability and allows the model to reach ~100 years per day simulated on the JOLIOT-CURIE French supercomputer (Sarr et al., 2019; Sepulchre et al., 2019). We nonetheless first provide a validation of the model on modern climatic conditions for the Asian monsoon regions.

We evaluate IPSL-CM5A2 ability to reproduce the climate patterns over Asia by comparing the last 20 years of a 1855-2005 historical run (Sepulchre et al., 2019) to the Global Precipitation Climatology Project (GPCP) for rainfall, and to the European Centre for Medium-Range Weather Forecasts (ECMWF) reanalysis (ERA-40) for the winds (Frauenfeld, 2005). Regarding precipitation, IPSL-CM5A2 shows typical biases shared with CMIP5-generation models, i.e. a ca. 2-month delay in the monsoon onset over India (see Supplementary materials, Figure 1), an underestimated extension of the monsoon over eastern China, Korea, and Japan, and an overestimation of rainfall rates over the subtropical western/central Pacific Ocean and Indian ocean (Sperber et al., 2013). However, these biases are reduced in IPSL-CM5A2 compared to the previous IPSL ESM version (Dufresne et al., 2013), as a response to a tuning-induced better SST pattern over the Arabian Sea that enhances rainfall over India during the summer monsoon (Levine et al., 2013).

Simulated mean annual precipitation (MAP) rates fits the main patterns of GPCP (Figure 1-a,b), although IPSL-CM5A2 tends to expand aridity over Arabia and Central Asia. Rainfall amounts over Nepal and Bangladesh are underestimated, whereas they are reinforced over the foothills of the Himalayas, likely as a response to the lack of spatial resolution that prevents representing orographic rainfall associated with the steep changes in topography of these regions. The expression of seasonality, calculated through the ratio of the precipitations during the 3 consecutive wettest month against the 3 consecutive driest month (hereafter 3W/3D) is well represented over Asia (Figure 1-c,d). We have chosen the 3W/3D ratio among many available criteria to characterize the climate seasonality because it has also been used as an indicator

of monsoonal climates (with a minimum threshold value close to 5) in previous investigations of paleo-monsoons (Herman et al., 2017; Shukla et al., 2014; Sorrel et al., 2017). The modern monsoonal regions in our model are adequately characterized by a high 3W/3D ratio, although this signature is stronger in southern Asia than in eastern China. The seasonality in precipitation is thus consistently reproduced for the modern.

Regarding atmospheric large-scale dynamics, winter monsoon winds are well simulated, with anticyclonic winds around the Siberian high (Figure 2-a,b). The summer circulation patterns (Figure 2-c,d) are also well reproduced, although the low pressure belt over Arabia and southern Asia is simulated with lower intensity and lesser extension than in the reanalysis. Likewise, the simulated EAM intrusion in eastern China appears to be less pervasive than in the reanalysis, in which winds coming from the South China Sea penetrate further inland. The simulated Somali Jet and SAM winds display weaker intensity, but mirrors the patterns observed in the reanalysis. Given that IPSL-CM5A2 reproduces well the seasonal atmospheric dynamics patterns and the seasonality, we will thus mostly focus on these criteria in the discussion that follows.

**2.2. Late Eocene fully coupled simulation set up**

The Eocene simulation (EOC4X) uses a 40 Ma paleogeography and paleobathymetry reconstruction (see Supplementary materials, Figure 2-a). Global plate reconstructions follow methods and plate references described in Baatsen et al., (2016) with significant modifications in tectonically active area within the 45-35 Ma based on a review of geologic data and literature (https://map.paleoenvironment.eu/). Specifically in the India-Asia collision zone, paleopositions are based on paleomagnetic references (Lippert et al., 2014) and collision is underway with greater India completely emerged (Najman et al., 2010). Based on a review of geological constraints (see Botsyun et al., (2019); Kapp and DeCelles, (2019) and references therein), the Tibetan Plateau altitude is set to 3500 m in Central Tibet forming a high elevated low-relief plateau. Moderate-low-elevation paleosurface for Northern Tibet and low-elevated regions further north into the Qaidam and Tarim Basins (surrounded by very subdued topography below 1000 m for the mountain belts of the Pamir, Kunlun Shan, Tian Shan, Altyn Shan and Qilian Shan / Nan Shan) decrease finally into the plain and epicontinental sea of Central Asia. The Parathethys is set to its extent estimated during the maximum ingression reached just before 41 Ma (Bosboom et al., 2017) and the Turgai strait, which connected the Parathethys sea and the Arctic Ocean, is set closed by mid Eocene (Akhmets'ev and Beniamovski, 2006; Kaya et al., 2019), but the water exchanges with the Tethys ocean are maintained to the south.

The $CO_2$ atmospheric concentration is set to 1120 ppm (4 PAL or 4X interchangeably), which corresponds to the high end of middle to late Eocene (42 - 34 Ma) $pCO_2$ estimates from data and carbon cycle models (Anagnostou et al., 2016; Beerling and Royer, 2011; Lefebvre et al., 2013). Ice sheets are removed as the presence of even small permanent ice sheets was highly unlikely under these $CO_2$ concentrations (DeConto and Pollard, 2003; Gasson et al., 2014). As other greenhouses gases ($CH_4$, $NO_2$, $O_3$) concentrations are poorly constrained for this period, they are left to their preindustrial values, as proposed in model intercomparison projects on pre-Quaternary periods (Lunt et al., 2017). The solar constant is reduced to 1360.19 $W/m^2$ (Gough, 1981) and the orbital parameters are set to their present values.

Although several vegetation reconstructions are proposed in the literature for the Eocene, they were usually designed for higher $CO_2$ concentrations (e.g. 8 PAL, in Herold et al., 2014) and/or different paleogeographies, such as the early Eocene (Sewall et al., 2000). Here, our Eocene fully coupled simulation uses an idealized vegetation map derived from the main

modern climatic zones on the globe (see Supplementary materials, Figure 2-b). The limits of this approach will be
discussed in the Discussion section.
**3. Results**
We first compare the simulated oceanic and terrestrial temperatures to two compilations of SST and mean annual
terrestrial temperatures (MAT), ranging respectively from 42 to 38 Ma (late-middle Eocene) and to 38 to 34 Ma (late
Eocene, a complete description of the compilation is given in the supplementary materials, Table 1 to 4). The main
climatic patterns over Asia obtained for EOC4X simulation are then presented and compared to the modern, and we
discuss potential implications on our understanding of the Cenozoic monsoon history.
**3.1. Comparison of the simulated Eocene climate with a proxy compilation**
The EOC4X ocean is initialized from warm idealized conditions similar to that proposed by (Lunt et al., 2017) and has
been run for 3000 years. At the end of the integration the ocean has reached quasi-equilibrium, including in the deep
oceanic layers, showing a drift inferior to 0.05 °C per century (see Supplementary Materials Figure 3). Our reference
simulation yields SST in better agreement with the 42-38 Ma late-middle Eocene group than with the late Eocene group
(Figure 4 in Supplementary materials). This suggests that our 4 PAL results are more representative of the late-middle
Eocene conditions, which seems consistent given the fact that 4 PAL corresponds usually to the higher $CO_2$ estimates for
the second half of the Eocene. Consequently, we develop here the comparison with the late-middle Eocene proxy group
(Figure 3), and attach the comparison between model and late Eocene proxy group (Supplementary Materials, Figure 5).
The comparison with SST estimates yields overall good results, although some discrepancies remain: at high latitudes,
DSDP 277 near New Zealand and ODP 913 in the North Atlantic show temperatures warmer by ~13°C compared to the
model, while in the Gulf of Mexico, the proxy is 11°C cooler than the model and in the equatorial Atlantic (site ODP 925)
proxies are 8°C cooler than the model. Despite a steeper latitudinal thermal gradient than that reconstructed from proxy
records, the model is able to match reasonably well the coldest and warmest proxy values (respectively 8° for the ACEX
drilling, in the Arctic and 36°C for JavaKW01 on the equator) with a +/- 3.5°C accuracy. This conundrum, where models
struggle to reproduce the flatter thermal gradient suggested by proxy records by simulating too warm (resp. cold)
temperatures at the equator (resp. poles), is a recurrent problem in modeling studies. Underlying causes remain unclear
and could be attributed to proxy uncertainties, missing processes in the models, (Huber and Caballero, 2011) or biases in
the way models handle small-scale processes, such as cloud feedbacks (Zhu et al., 2019). Regarding the proxies, seasonal
bias (towards summer or winter, Schouten et al., 2013; Tierney and Tingley, 2018) might affect the temperatures
interpreted as representative of the mean annual sea surface temperature. Calibration methods can also be questioned,
especially for warmer than present deep-time studies, as it the case for Mg/Ca paleothermometer that doesn't account for
the changing Mg/Ca ratio of seawater (Tierney et al., 2019), or for the $U^{K'}_{37}$, for which recent BAYSPLINE recalibration
method have proven to lower the mismatch at high temperatures (Tierney and Tingley, 2018). In the case of the $TEX_{86}$,
a subsurface bias has been suggested (Ho and Laepple, 2016) and remains debated (Tierney et al., 2017). In Asia, $\delta^{18}O$
measurements in oyster shells from the eastern edge of the Paratethys sea spanning the second half of the Eocene give
estimates for the mean annual temperature as well as the seasonal amplitude, yielding SST estimates ranging from 22°C

in winter to 38°C in summer (Bougeois et al., 2018). The simulated SSTs are consistent with these values, with a coldest simulated SST of 15°C in January and a warmest simulated SST of 35°C in August.

The fit between modeled and terrestrial proxies MAT (Figure 3-c,d) is less successful. The model reasonably fits temperatures in Australia, South America, Antarctica, Greenland and Europe with a mismatch between values staying below +/-5°C for all locations, except the Gran Barranca (Chile) and Stare Seldo I (Europe) points. On the other hand, larger differences exist over North America and Asia, although the mismatch might likely have different origins. All of North American proxy sites are located close to the West coast and to the Rocky Mountains, the Cenozoic history of which is also complex. Incorrect prescribed topography in the model as well as local effects of atmospheric circulation might therefore have a large impact in terms of reconstructed temperatures (Feng and Poulsen, 2016; Sewall and Sloan, 2006). We note that the model successfully represents the proxy temperature range in this region (between 3 and 23°C for the proxies and between 4 and 27°C for the model), which suggests that the model-data mismatch is more likely related to paleoelevation errors or local climatic effects rather than to a systematic bias in either the model or proxies. On the contrary, the remarkable homogeneity amongst the estimated MAT from Asian proxy records (ranging only from 14 to 19°C) is somewhat puzzling, considering the fact that these 28 sites are spread between 18 and 52°N in latitude and are located in various geographical settings, ranging from coastal regions to mountainous areas. A possible cause could be the application of modern temperature-vegetation relationships to paleobotanical records, which might not prove fully adequate to reconstruct the warmer climates of the Eocene (Grimm and Potts, 2016; Peppe et al., 2011).

If quantitative comparisons between model and paleovegetation data need to be treated with caution for climates warmer than modern, fossilized plants, together with lithological proxies, do however provide useful qualitative information. In Asia, Eocene proxy reconstructions converge towards a generally zonal climatic pattern, with a dry arid belt spreading from the Tarim basin to the east coast of China (Sun and Wang, 2005), and fringed by more humid climates over India and South East Asia on its southern flank (Licht et al., 2014; Ma et al., 2012; Sun and Wang, 2005) and over Siberia to the North (Akhmetiev and Zaporozhets, 2014). In the next sections, we will focus on the atmospheric circulation simulated for our Eocene simulation and analyze the shape and occurrence of the different Asian monsoons.

**3.2. Asian Eocene atmospheric circulation**

EOC4X seasonal atmospheric circulation patterns are presented for winter (December-January-February) (Figure 4-a) and summer (June-July-August) (Figure 4-c) and compared to their modern counterparts (Figure 4-b,d). The winter circulation is characterized by a strong high-pressure belt at latitudes lower than today, located over the proto Himalayan Tibetan Plateau between 20 and 45°N. Strong westerlies are simulated at mid-latitudes around 40-50°N and easterlies at latitudes lower than 20°N (up to 15 m/s against 5 m/s in the Control simulation). These features contrast with the modern winter system characterized by zonal winds with a lower intensity and a larger meridional component. Finally, no analogue to the modern Siberian High is simulated at 40 Ma (Figure 4-b). Today, the Siberian High is controlled by winter surface temperatures dropping below the freezing point in northeastern Siberia (around 50°N). In our Eocene simulation, the combined effect of a warmer climate and a reduced continentality (due to the presence of the Paratethys and Siberian seas) prevent its development.

During summer months, the nearby presence of the Tethys ocean and Paratethys sea results in a large high-pressure cell centered over 30°E and extending from 10° to 50°N (Figure 4-c). The Tethysian high is associated with intense 850 mb northerlies around 60°E which are partly deviated into northwesterlies when sweeping over northern Greater India (Figure

4-c). To the south, 850 mb winds originated from the Indian Ocean enter the Indian subcontinent at low latitudes (<10°N)
and turn southeasterlies over the Bengal Bay to feed precipitations over the foothills of Himalaya before shifting to
southwesterlies (Figure 4-c). In the modern configuration, the 850 mb winds of the SAM originate from the Indian Ocean
and extend northward up to 20°N over India before taking a northeast direction and generate heavy precipitations from
India to Myanmar and up to the southern flank of the Himalayas to the North (Figure 4-d, Figure 5-d). These precipitations
over southern Asia (up to 15 mm/day, Figure 5-d) are fed by the Somali Jet, a strong low-level cross-equatorial moisture
flow originating from the Indian Ocean which turns anticyclonically in the northern hemisphere along the eastern edge
of the eastern African relief (Figure 4-d).
Figure 5 shows the northward moisture transport vertically integrated over the whole atmosphere column for the Control
and EOC4X experiments. In the Control Experiment, the largest meridional moisture transport crossing the Equator is
simulated along the Eastern African coastline (Figure 5-b) and corresponds to the strongest meridional wind component.
It confirms that the Somali Jet is a key feature of the modern Southern Asian Monsoon (Figure 5 b,d). Conversely in the
EOC4X experiment, the Somali Jet (0-10°N/45-50°E) barely exists. Instead, moisture flows from the Tethys and Indian
Oceans towards western Africa, where heavy summer precipitations are simulated (over 30 mm/day, Figure 5 c). This
alternate moisture pathway toward western Africa rather than southern Asia is probably the result of several
paleogeography features (African continent positioned farther south, absence of topography in eastern Africa, presence
of a Tethys seaway preventing the south Asian low pressure to extend westward) and will be discussed further in Section

282    4.2.

In the western Indian ocean, the cross-equatorial moisture flow is strongly reduced in EOC4X compared to the Control
simulation, whereas it is increased over the eastern Indian ocean. However, this diverted equatorial moisture flux remains
below 10°N and the Asian eastern Pacific coast receives instead a mixture of westerly winds coming from northern India
(above 30°N) and weak easterly winds bringing moisture from the Pacific Ocean at lower latitudes (Figure 5-c),
contrasting strongly with the modern EAM (Figure 5-d).
These atmospheric changes, both in summer and winter, generate a large arid area extending throughout western China,
the proto-Tibetan Plateau and northern India, while southern India and Myanmar experience intense rainfall due to their
position closer to the equator in the Eocene (Figure 6-a,b). Apart from changes in near surface winds, two intertwined
processes conspire to explain the aridity: (1) a rise in the water vapor condensation height (corresponding roughly to the
cloud base) and (2) a weakly convective atmospheric column. The first process arises from the extreme surface air
temperature in EOC4X (up to 45°C), which results in a simulated water condensation altitude that exceeds 3500 m over
Northern India and Tibet. This altitude corresponds to a pressure level of ~680 mb (in the middle troposphere), while the
water condensation altitude remains below 2500 m in the control experiment, which corresponds to a pressure level of
~800 mb (in the lower troposphere, Figure 6-c,d). The second process, the lack of deep convection, makes mid-level
atmospheric layers very dry and prevents air masses to reach the water condensation altitude, as shown by two longitude-
altitude cross sections of the relative humidity at 20°N and at 40°N (Figure 7).
At 20°N today, modern India and Southeast Asia show multiple deep convection centers and a relative humidity around
60% in most of the troposphere (Figure 7-d). In contrast, the Eocene displays a more stratified atmosphere, with two weak
convective cells above the Indian and Southeastern Asian land masses, which are blocked around 600 mb by subsiding
air masses. Locations of deep convective heating can also be highlighted by observing the upper troposphere temperature
maxima in the tropics (Boos and Kuang, 2010; Privé and Plumb, 2007; Roe et al., 2016), as presented in Figure 8. In the
Control experiment, upper temperature maxima are located over northern India deep convection regions (Figure 8-b),

which is in good agreement with reanalysis (see SI, Figure 6). Deep convection tends to occur where latent and sensible heats per unit mass maximize which is close to the subcloud surface (Emanuel et al., 1994), where temperature and relative humidity are elevated. In the control experiment, deep convection over India appears to be mostly controlled by latent heat because evaporation of precipitated water ensures moisture availability. Yet, in EOC4X, the latent heat over India is largely weaker due to a lack of moisture despite warmer temperatures. Consequently no upper-level temperature peak is simulated over northern India but rather over the Western Pacific (Figure 8-a), where both temperature and relative humidity are the highest.

At 40°N, the presence of the Paratethys sea and the Tarim basin as far as 80°E is translated into a shallow surface of high relative humidity (~70%, see Figure 7-a), which is confined in the lowest troposphere levels by strong subsiding winds. The deep convection is here again muted by large-scale mid-level atmospheric dynamics. These diagnostics converge to demonstrate that our simulated Eocene atmosphere in Asia has little in common with the modern. The application of the Webster and Yang Index (WYI) (Webster and Yang, 1992) further confirms these atmospheric contrasts. The WYI is a standard diagnostic criterion for the SAM that quantifies the shear effect between the lower and higher troposphere, which is a typical characteristic of this monsoon. Modern WYI summer values over the northern Indian Ocean exceeds 20 whereas our EOC4X simulation yields summer values below 6 (Figure 9), thereby emphasizing the strong differences between Eocene and modern summer circulation patterns in this region.

**4. Discussion**

**4.1. Can proxies identify monsoons?**

The comparison of our model results showing a broad arid zone over Asia, with late Eocene proxy records is reasonably good despite the fact that many of these records have been used to infer the existence of monsoons. This is first shown by a simple qualitative comparison with vegetation reconstructions from the Middle Eocene (Figure 6-a), derived from a compilation of paleobotanical studies (detailed in supplementary materials). The spatial distribution of forests and shrubland/grassland inferred from these studies is mostly coherent at first order with simulated MAP, however, a discrepancy remains between the northern Indian and Bengal forests and the dry conditions simulated (< 1mm/day). This could be attributed either to a bias towards aridity in these specific regions, that is shared by most models (Valdes et al., 2017) and seems to translate in the Eocene as well, and/or to an inaccurate reconstruction of northern Indian late Eocene topography. We have indeed shown that, although our model reasonably simulates the modern monsoons in a control simulation in terms of wind regimes, the amount of precipitations simulated is underestimated, especially in India and in the Bengal region (Figure 1-a,b). This, together with the large error bars associated with most of the quantitative reconstructions on precipitations proposed by paleobotanical studies, hampers a quantitative comparison to paleovegetation records, which mostly provide estimates of mean annual precipitation amounts. We thus rather focus on Eocene proxy records of seasonality (as previously done in Huber and Goldner, 2012), for example) as of our model's ability to produce seasonality metrics in good agreement with modern observations (Figure 1-c,d).

Figure 8 shows the 3W/3D obtained with EOC4X and compared to the Late-Middle Eocene compilation of coal and evaporites deposits from Boucot et al., (2013). In the literature, evaporites are traditionally interpreted as markers of seasonal to arid environment, while coals indicate more stably wet climates, and thus have been extensively relied on to infer past climates (Huber and Goldner, 2012; Sun and Wang, 2005; Ziegler et al., 2003). However, this approach has

been criticized as oversimplistic (Wang et al., 2013; Wiliams, 2007). Therefore, in addition to this compilation, we highlighted localities positioned in strategic regions and resulting from robust multi-proxy analysis, that were recently used to suggest monsoon-like highly seasonal climatic conditions during the late-middle Eocene (Figure 10-a): 1) the Tarim region (Bougeois et al., 2018); 2) the Xining Basin, located at the interface between the zones of influence of the modern westerlies and of the EAM (Meijer et al., 2019); 3) the Maoming/Changchang basins in southeastern China (Herman et al., 2017; Spicer et al., 2016), located in the transition zone between EAM and I-AM; 4) the Jiuziyan Formation (Sorrel et al., 2017) and finally 5) the Pondaung formation in Myanmar (Licht et al., 2014), presently located in the area of influence of the SAM. Although we lack Indian sites suggesting the presence of the SAM in the late Eocene, we acknowledge that such sites do exist for the early Eocene (i.e. the Guhra mine in Rajasthan, Shukla et al., (2014).

When compared to our model results, most of the evaporite deposits and highlighted localities are found in regions of strong seasonality (3W/3D > 5, purple outline in Figure 10-a), except for the Myanmar site located in a more ever-wet context and the Tarim region, which experiences a mostly ever-dry climatic context. As many of these highlighted localities stand on the edge of our simulated arid zone, we suggest that the extension of this region might be modulated by orbital forcing, as both models and data seem to suggest (Abels et al., 2011; Licht et al., 2014; Sloan and Morrill, 1998; Zhang et al., 2012), which should be the topic of further investigations. Inversely, most coal bearing deposits stand in regions of very low seasonality and relatively high MAP (southern India, southern Myanmar, northeastern China), although some discrepancies remain in northern India and Bengal regions, which could be linked to the aforementioned dry bias of the model and/or to regional bias induced by specific coal depositional environment. The comparison of coal and evaporites deposits to late Eocene MAP, although less reliable for the reasons mentioned above, follow a comparable pattern, as most of coals settle in regions of relatively high MAP (> 1000 mm/year) while the evaporites, on the other hand, are present in drier locations (SI, Figure 7).

These results, together with the previously shown wind patterns highlight that Eocene seasonality and wind regimes might have been substantially different from the modern conditions. We argue that high seasonality criteria (3W/3D or similar) may equally result from either SAM, EAM, or ITCZ seasonality (WNPM or I-AM), and therefore hardly discriminate between these different mechanisms. This ambiguity is also apparent in the proxy records. For example, markers of highly seasonal precipitations found in Myanmar were successively interpreted as indicators of a modern-like SAM (Licht et al., 2014), then to a migrating ITCZ-driven monsoonal rainfall due to revised paleolatitude of the Burma terrain (Westerweel et al., 2019). Additional seasonality data in targeted areas as well as the application of new techniques on fossil leaves (Spicer et al., 2016) that are promising in their ability to distinguish between the different seasonal signals (ITCZ, SAM, EAM) might in this regard bring meaningful insights on new and existing sites and together with modeling results help resolve the question of the monsoons initiation timing.

### 4.2. What drove the inception of Asian Monsoons?

The atmosphere dynamics over Asia in our Eocene simulation presents significant differences relative to the modern. It indicates the existence of a latitudinal extensive arid zone over northern India and central Asia bordered by areas of highly seasonal precipitation, however our results do not produce monsoonal circulations in the modern sense. The absence of a true paleo-monsoon contrasts with the findings reported in some previous Eocene modeling studies but a large arid zone is consistent with other model studies of Eocene or other time periods as detailed below. This interestingly suggests that the boundary conditions necessary for the inception of monsoon-like circulations may have occurred within this broad

greenhouse timeframe and, more importantly, that the monsoon-triggering conditions may be determined by comparing these various model studies with our results and proxy data. Indeed, each study has its own modeling setup and differences in the results might come from either the choice of model, the model resolution and/or the boundary conditions that were used. If all the CMIP5 generation models, except for CCSM4, experience the same dry bias in Asia when compared to modern observations (Valdes et al., 2017), and if a better resolution appears to have limited impact on the outcoming results (Huber and Goldner, 2012; Li et al., 2018), the paleogeography is a key point to consider. Indeed, recent studies suggest that paleogeography is the key driver shaping eastern Asian climate (Farnsworth et al., 2019) and (Lunt et al., 2016) further showed that paleogeographic changes observed during the Eocene could be responsible for mean annual temperature changes that might be as high as +/- 6°C.

Several main diverging paleogeographic characteristics stand out between all the available modelling studies regarding the Eocene. First, the position of the Indian continent, which either is fully disconnected from Asia and in an equatorial position (Huber and Goldner, 2012; Zhang et al., 2012) or has already collided with Asia (this study; Li et al., 2018; Licht et al., 2014; Zhang et al., 2018). Second, the orientation and the latitude of HTP significantly differ from a study to another (Huber and Goldner, 2012; Licht et al., 2014; Zhang et al., 2018). Third, the Turgai strait that is either represented as open (Li et al., 2018; Licht et al., 2014; Zhang et al., 2012) or close (this study, Huber and Goldner, 2012; Zhang et al., 2018). Fourth, the elevation of oriental Siberia, that displays variable elevation ranging from <1000m (this study, Huber and Goldner, 2012; Zhang et al., 2012) to more mountainous (1000 to 2000 m) configurations (Li et al., 2018; Licht et al., 2014; Zhang et al., 2018). Given that some of these key features of the late Eocene paleogeography are still highly debated (Kapp and DeCelles, 2019, for a review), we propose below a short review of previous studies and the possible impact of varying boundary conditions on resulting Asian climate.

There are competitive models for the evolution of the Indian Foreland seaway, with some predicting the presence of a deep sea between Continental India and Asia (Jagoutz et al., 2015; van Hinsbergen et al., 2012) or an epicontinental sea (DeCelles et al., 1998) in the early and middle Eocene. However, geological evidence indicates that the Indian Foreland seaway have dried out by 40 million years (Najman et al., 2008) and terrestrial connection is suggested even earlier, around 53.7 Ma, according to paleontological evidence based on mammalian fossils (Clementz et al., 2011). In that aspect, the existence of a seaway between India and Asia (Huber and Goldner, 2012; Zhang et al., 2012), is clearly representative of the early Eocene. Regardless of the exact timing for the complete emergence of Greater India, the presence of a seaway in these warm low latitudes certainly represents an important water vapor source to the surrounding regions (Tibetan Plateau, northern India, Bengal), and could therefore reduce the aridity of this area.

Interestingly, the Tethys/Paratethys region from Huber and Goldner, (2012) presents more similarities with the early Miocene, as northern Africa and Arabia are fully emerged while the remnants of the Paratethys sea in Europe are reduced to small inner seas. We hypothesize that the increased continentality in Europe and northern Africa in their experiment may contribute to prevent the formation of a Tethysian anticyclone (as simulated in the present study), hence generating atmospheric circulation more similar to the modern. On the other hand, also with the use of an early Eocene Indo-Asian configuration but a broader Tethys ocean and Paratethys sea, Zhang et al., (2012) obtain results that are more similar to ours in terms of sea level pressure and seasonal winds. This supports the importance of the Paratethys extension in shaping Asian climate, which was already suggested by previous studies (Fluteau et al., 1999; Ramstein et al., 1997; Zhang et al., 2007).

Recent findings suggest that the latitudinal position of the TP exert a control over Eocene Asian climate, especially summer wind patterns (Zhang et al., 2018) and therefore add another level of uncertainty given that the location and

elevation of the TP in the Eocene is still debated (Botsyun et al., 2019; Wang et al., 2014). When oriented in a NW-SE direction and located between 10 and 20°N, the TP blocks summer equatorial winds transporting moisture northward and enhances orographic precipitations over the southern flank of the TP, while westerly winds coming from the Paratethys cross central Asia without encountering major orographic barriers (Licht et al., 2014; Zhang et al., 2018). On the contrary, configurations with a modern TP position (between 30 and 40°N) deviate the westerlies coming from the Paratethys into a counter-clockwise flow around the southern flank of the TP (this study; Zhang et al., 2018).

The Turgai strait configuration and oriental Siberian paleotopography might also have a significant impact on the simulated Asian climate. An open Turgai Strait (Li et al., 2018; Licht et al., 2014; Zhang et al., 2012) maintains a connection between the warm Paratethys sea and the colder Northern sea and might result in an increased land-sea thermal gradient in the western Asian mid latitudes by cooling the Paratethys. Providing that the seaway is deep enough to allow for such heat exchanges, it could amplify the land-sea breeze phenomenon in summer between the Paratethys and the Asian continent and play a part in Central Asian water budget. A more mountainous configuration of inner Siberia (Li et al., 2018; Licht et al., 2014; Zhang et al., 2018) might generate colder winter temperatures and create the conditions required for the inception of a proto Siberian High in winter. Although the closure of the Turgai strait is estimated to occur around mid-Eocene (Akhmetiev and Zaporozhets, 2014), Siberian paleoelevation remains highly speculative and, to our knowledge, neither have been the topics of in-depth modelling studies.

In summary, current knowledge about late Eocene Asian paleogeography is not yet sufficient to discriminate between the various model solutions obtained with different boundary conditions. Moreover, some models come to the same conclusions using different paleogeographic reconstructions. This review, however, indicates and identifies potential paleogeographic boundary conditions have driven the shift from arid zonal Asia to Asian monsoonal conditions. We also argue that a modeling intercomparison project focusing on late Eocene Asian climate, using similar boundary conditions and applying similar diagnostic criteria, would definitely be a valuable asset to the community to provide a consistent picture of the initiation and evolution of Asian monsoons from a modeling perspective.

## 5. Conclusion

The earth system model IPSL-CM5A2 is able to catch modern Asian main climatic features and to produce Eocene climatic reconstructions which seem realistic when compared to proxy SST estimates and are comparable to recent studies that proposed a global climate reconstruction for this time period using earth system models (Baatsen et al., 2018; Hutchinson et al., 2018; Inglis et al., 2015). Our results point out notable differences in terms of wind patterns and precipitation amounts in Asia when compared to modern circulation, suggesting that no SAM neither EAM were occurring at that time, although highly seasonal climate is modelled in these regions. Our climate simulation rather proposes the existence of a wide arid zone in northern India and central Asia, due to the presence of strong subsiding winds in the mid troposphere, preventing the moist air coming from the equator to condensate and precipitate over the continent. These simulations suggest that these conditions prevailed before the set-up of the modern SAM and EAM, more likely appearing after the late Eocene, by contrast to what is found in other simulations (e.g. Huber and Goldner, 2012). If the existence of this arid climate is closely linked to the late Eocene paleogeography, the scarcity of paleo data in this simulated arid region remains a limitation.

We suggest that investigating the precise period when Asia transitioned from arid zonal climate to modern-like monsoonal climate would require collecting data in this specific arid area. Ultimately, we believe that additional simulations

performed using different models forced by identical boundary conditions as well as new Paleogene records from Asia (especially in southeastern Asia and India) are needed to draw more precise conclusions on the appearance of Asian monsoons and their potential existence in that period. Also, more recently developed modelling techniques could be very promisingly applied as a complement to complex climatic modelling reconstructions. For example, isotopic-enabled models, by simulating paleoprecipitations $\delta^{18}O$, allow a direct comparison of the model output to $\delta^{18}O$ values that can be measured in a wide variety of proxies (shells, carbonates, etc.) and therefore provide robust physical mechanisms to explain the measured patterns (Botsyun et al., 2019; Brady et al., 2019; Poulsen et al., 2010; Risi et al., 2010). Additionally, the application of proxy forward modelling methods (Evans et al., 2013; Schmidt, 1999), by mimicking the mechanisms through which a particular proxy will record a climatic perturbation (e.g. the translation of water $\delta^{18}O$ variations by planktonic foraminifera) taking into account the proxy's specificity (e.g. ecology of the foraminifera, episodes of secondary calcification and dissolution) and the time uncertainty could contribute greatly to help fill the gap between proxy records and model results.

471

## 6. Appendices: Figures

**Figure 1: Comparison of mean annual precipitations in mm/year (a,b) and 3wet/3dry ratio (c,d) simulated in the modern control simulation (a,c) and the GPCP observations (b,d).**

475

476

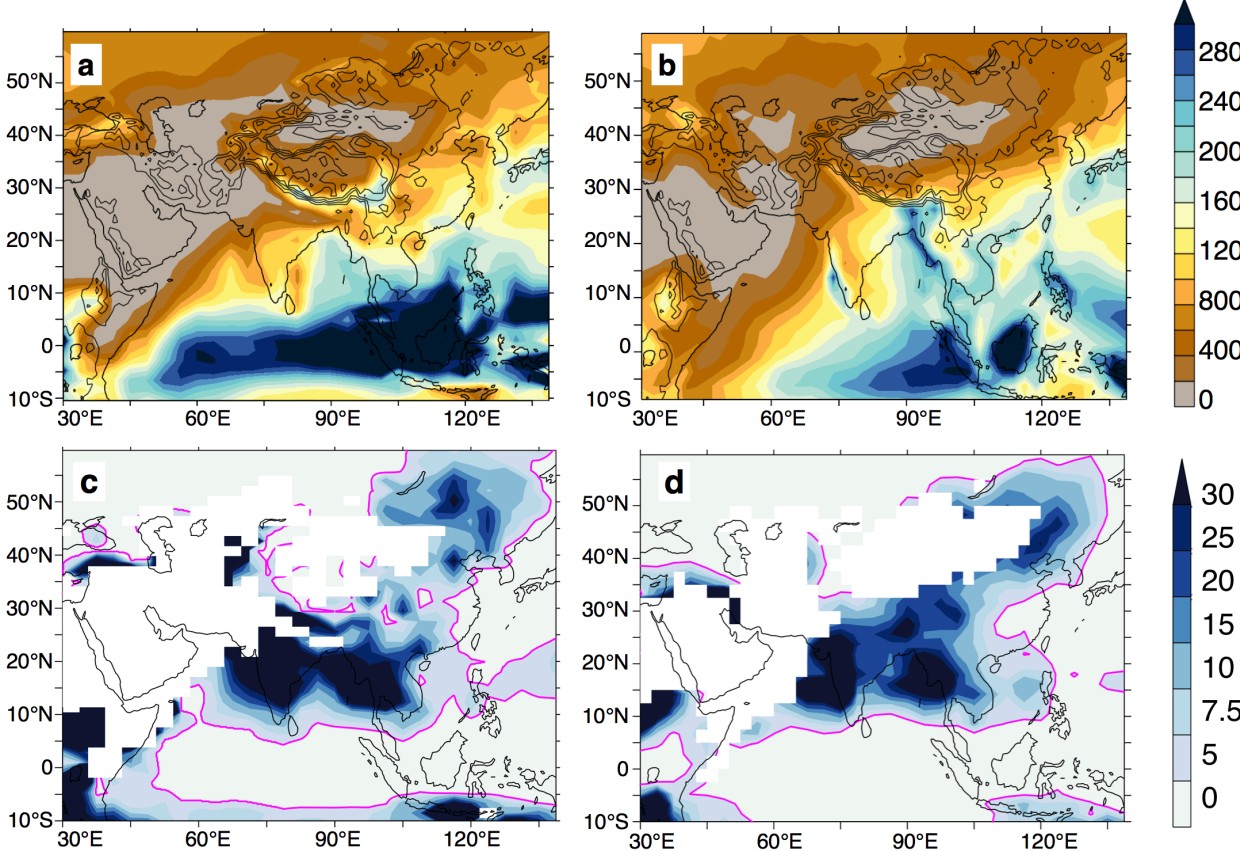

477

478

**Figure 2: Comparison of January to March (a,b), June to August (c,d) mean wind patterns obtained in the modern control simulation (a,c) with ERA40 reanalysis (b,d). Shading represents Sea Level Pressure anomaly (in mb), calculated as the difference between seasonal SLP minus the mean annual SLP. Overprinted vectors show 850 mb wind speed expressed in m/s. Main zones of high (low) pressure are highlighted with H (L) black letters. Main features of the summer monsoon are highlighted in red: Somali Jet (SJ), South Asian Monsoon (SAM) and East Asian Monsoon (EAM).**

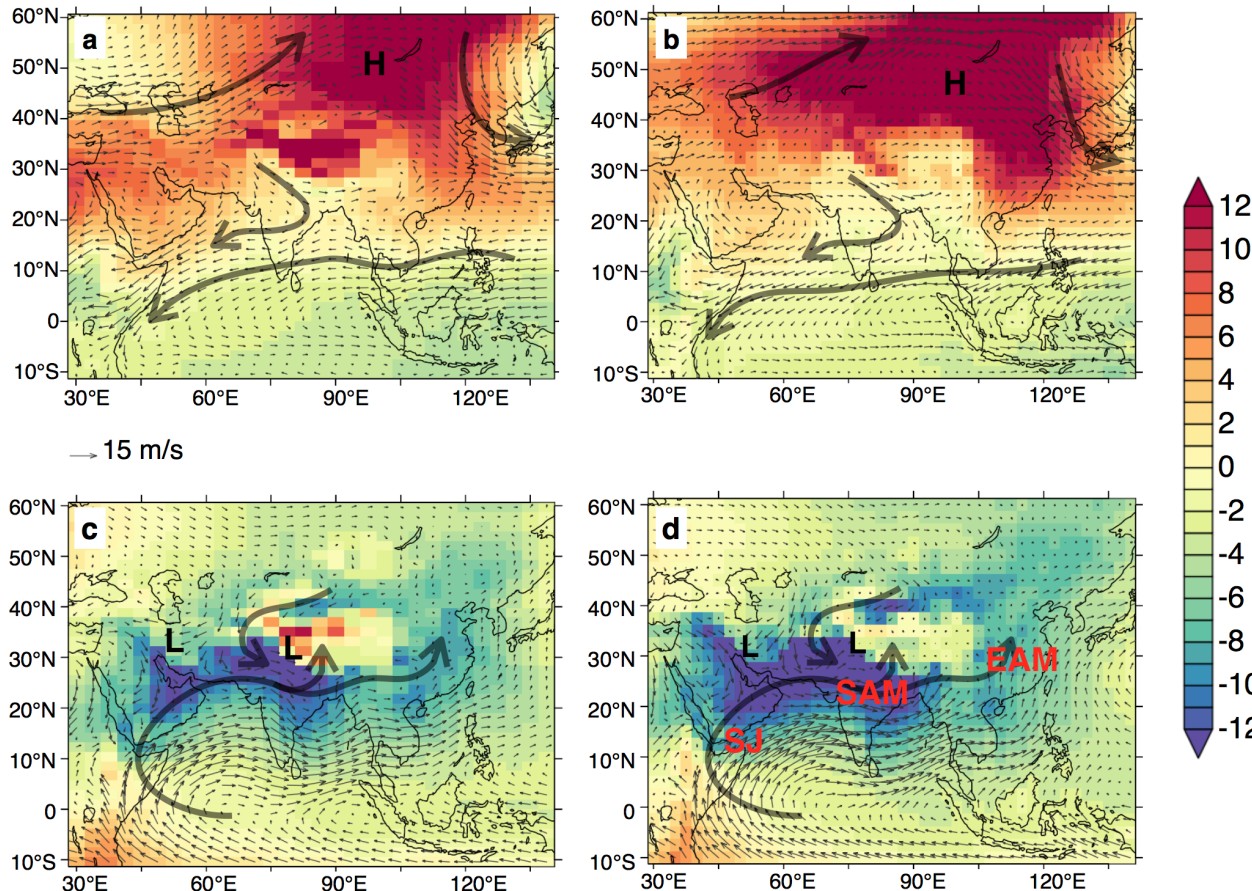

 **Figure 3: Late-middle Eocene global model-data comparison for SST (a,b) and MAT (c,d). In (a, c), thick line**
**represents the mean temperature from EOC4X, thin lines the min and max latitudinal temperatures from EOC4X.**
**For terrestrial proxies (d), high altitude locations (>1000 m) are represented by triangles, the others by circles and**
**pink thick line represent the 10°C isotherm.**

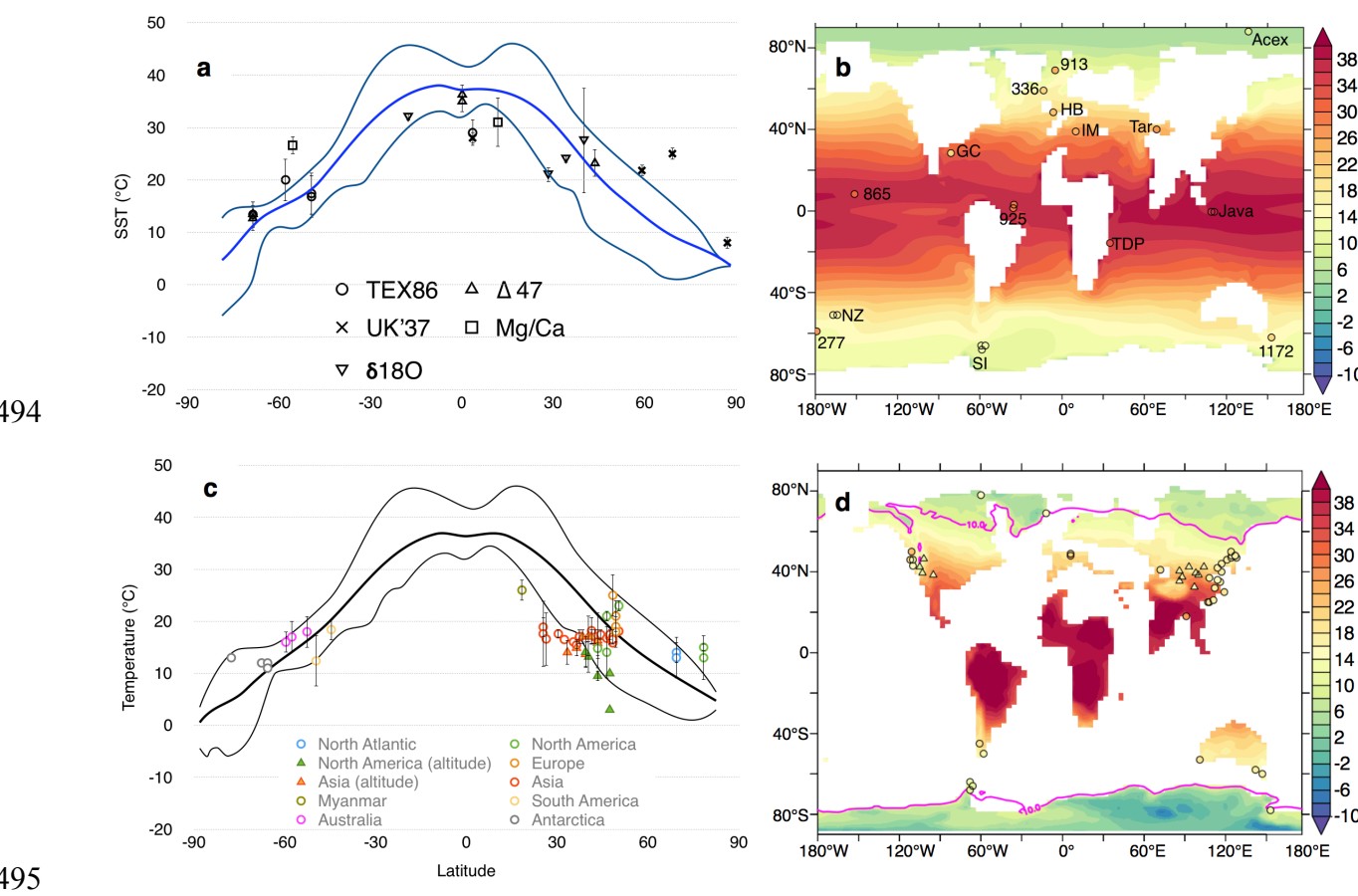

Figure 4: Sea level pressure anomaly (shading, in mb) and 850 mb wind patterns (vectors, m/s) obtained in EOC4X (a,c) and compared to control simulation results (b,c). (a,b) Winter circulation pattern; (c,d) summer circulation patterns. Main wind patterns are represented by a thick black arrow, and low pressure zones (high pressure) are marked by letter L (H). Numbers corresponds to regions highlighted in the main text: 1 Tarim sea region (Bougeois et al., 2018), 2 Xining Basin (Meijer et al., 2019, Licht et al., 2014), 3, Maoming Basin (Herman et al., 2017, Spicer et al., 2016, 2017), 4 Jiuchuan basin (Sorrel et al. 2017) and 5 Pondaung formation in Myanmar (Licht et al., 2014).

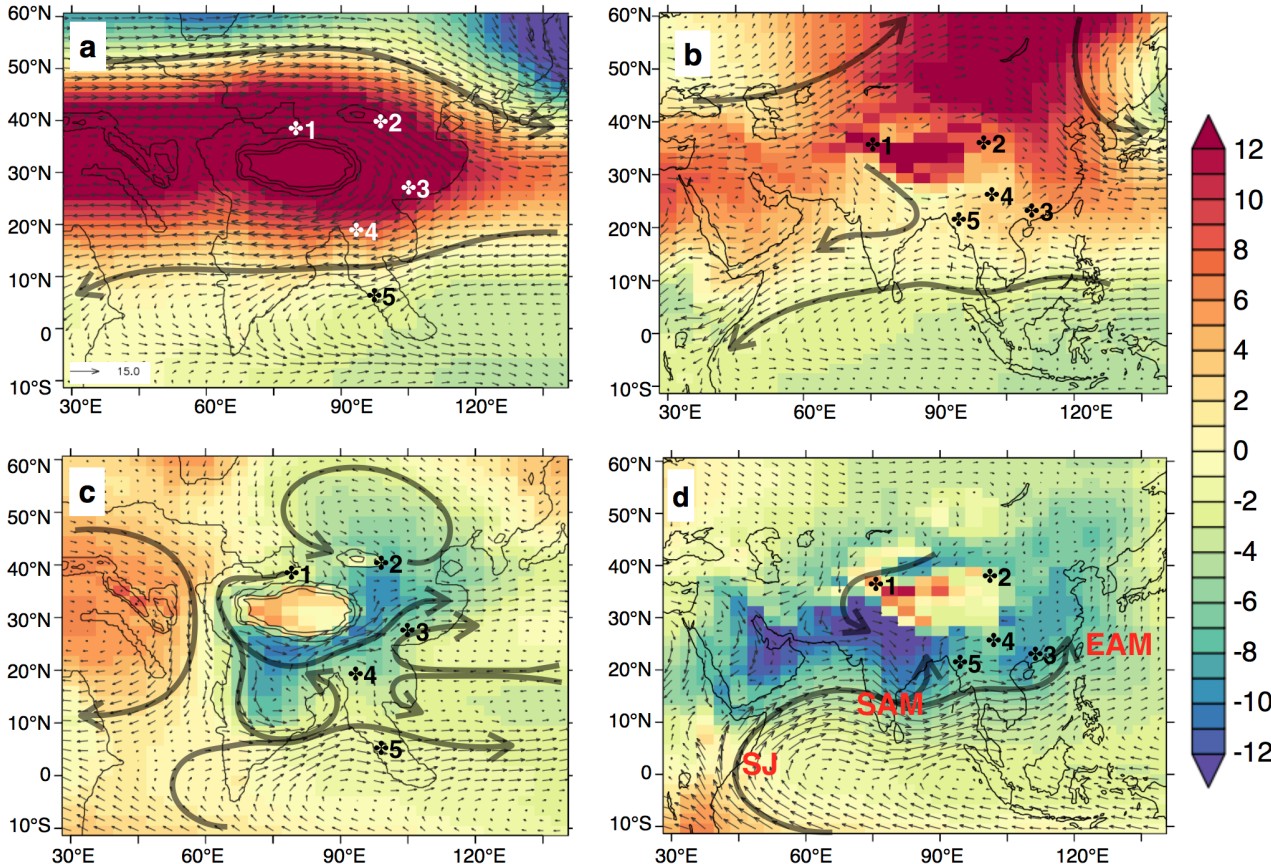

**Figure 5: Comparison of EOC4X (a, c) and Control (b, d) water column integrated moisture fluxes. (a, b) Vertically integrated northward JJA moisture transport averaged between 2°S and 2°N (black lines and left axis) and expressed as:**

$$\vec{Q} = -\frac{1}{g}\int_{p_s}^{0} q\vec{V}\,dp$$

**where q is the specific humidity and V is the horizontal wind vector, g the gravitational acceleration and ps the surface pressure; dotted lines represent the elevation of land masses within the same latitudinal band (right axis); arrows and legends indicate the direction of the zonal component of moisture fluxes. (c, d) JJA moisture fluxes (vectors) and cumulated precipitations for the same period (mm/day). Black boxes highlight the area used to compute meridional moisture fluxes in (a, b).**

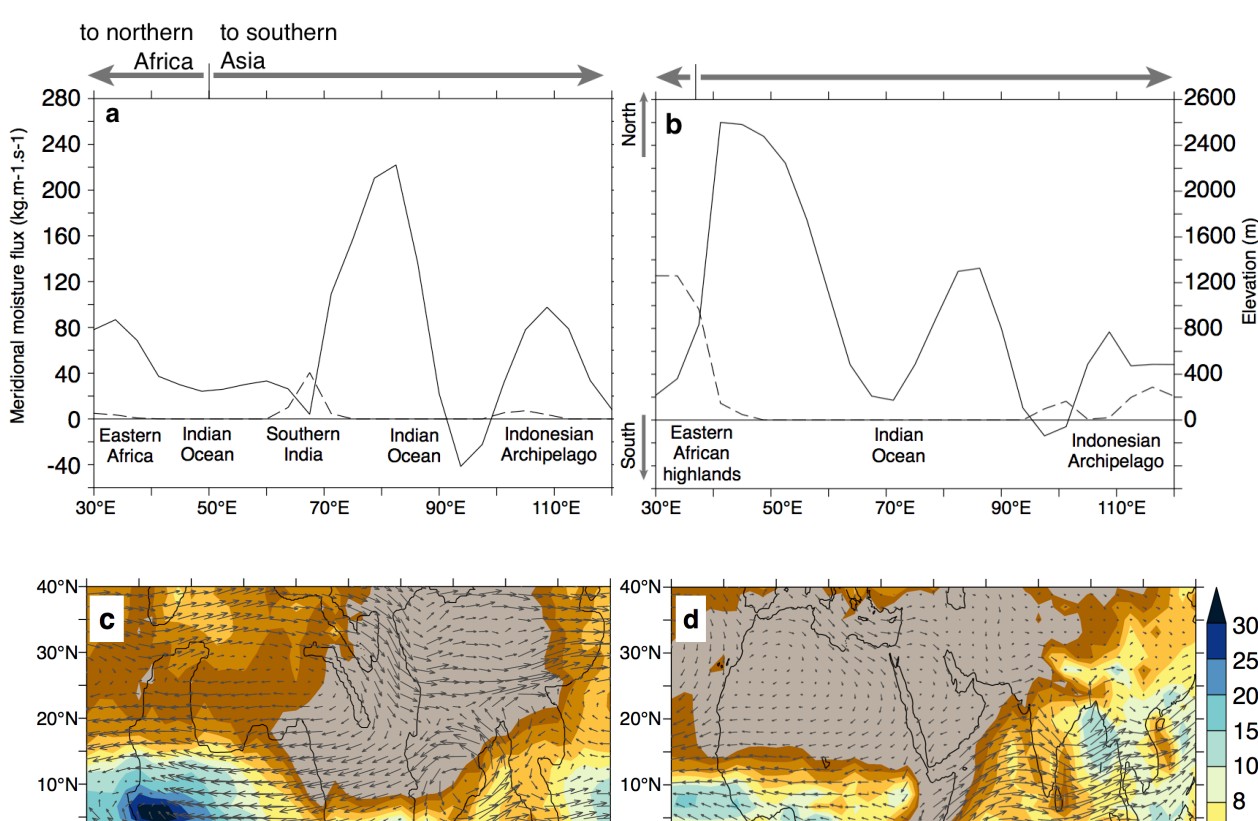

**Figure 6: (a,b) Mean annual precipitations (mm/year) for EOC4X simulation (a) and the Control simulation (b). The green outline delimits the arid region receiving less than 1mm/day. (c,d) Water condensation altitude (in m) in July for EOC4X simulation (c) and Control simulation (d). Horizontal dotted lines show the latitude used for the meridional profiles in Figure 5. In (a), circles indicate location of paleovegetation studies and describe forested environment (green) and shrub/grass environment (red), according to qualitative descriptions described in the Supplementary Materials (Table 5 ).**

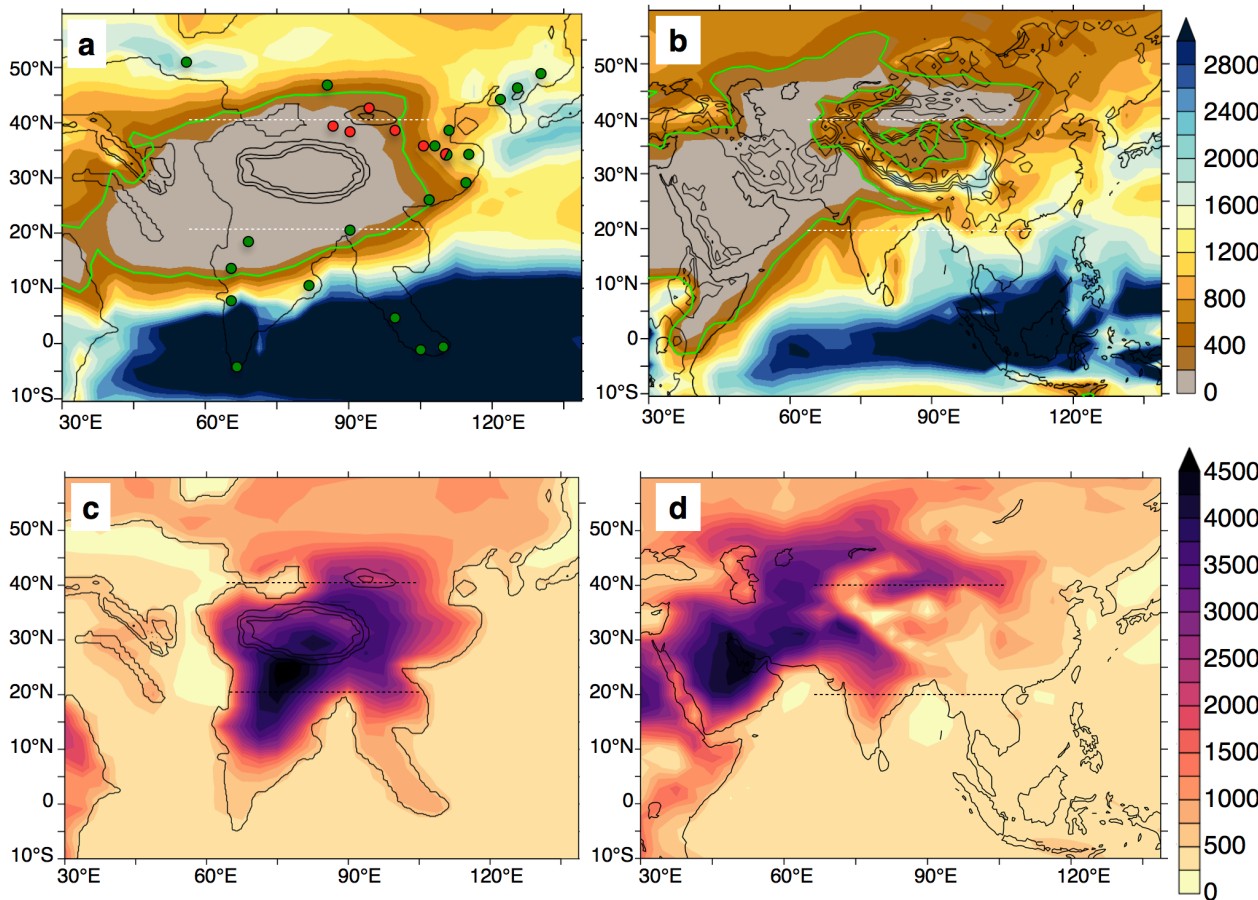

Figure 7: Longitude-Altitude profiles of the relative humidity (shaded) and vertical winds (vectors) for EOC4X (a,c) and control simulation (b,d), at 40° N and (a,b) and 20° N (c,d). Values are taken from the month of July.

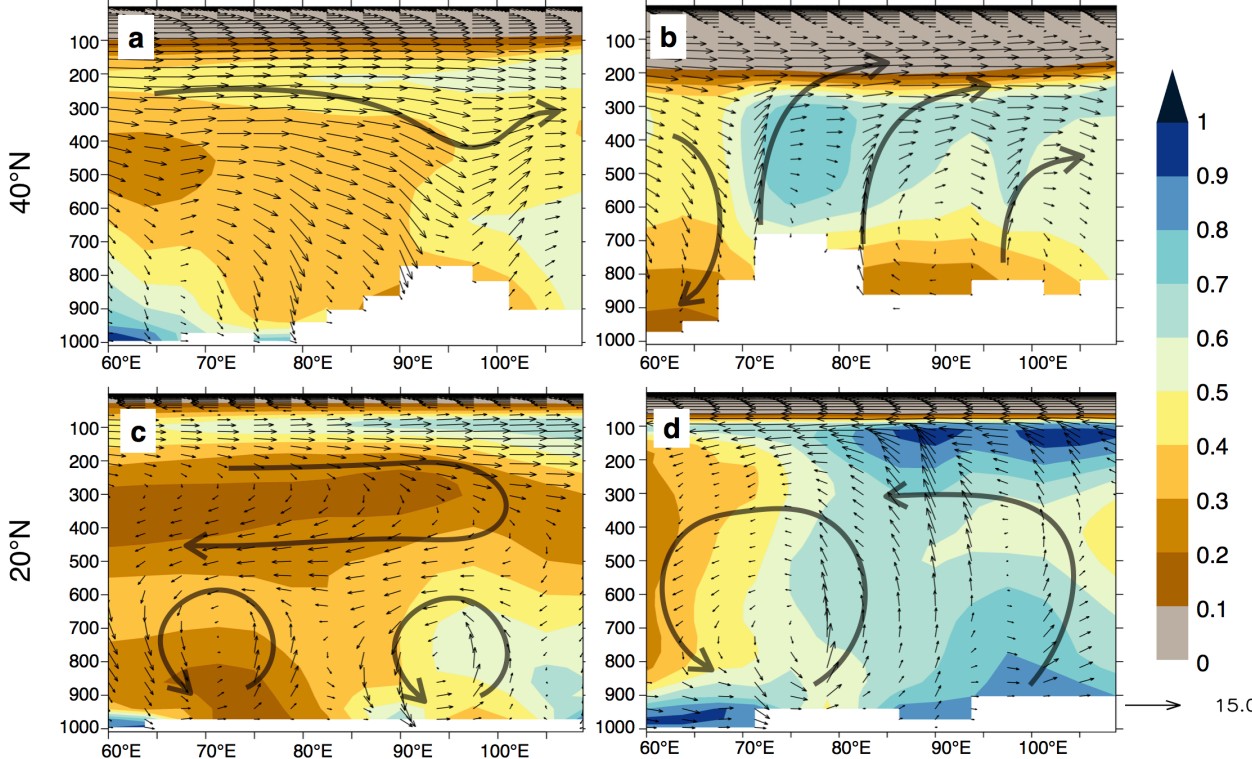

Figure 8: JJA Air Temperature (in Kelvin) at 300 mb for EOC4X (a) and Control (b) with contours overlaid each degree.

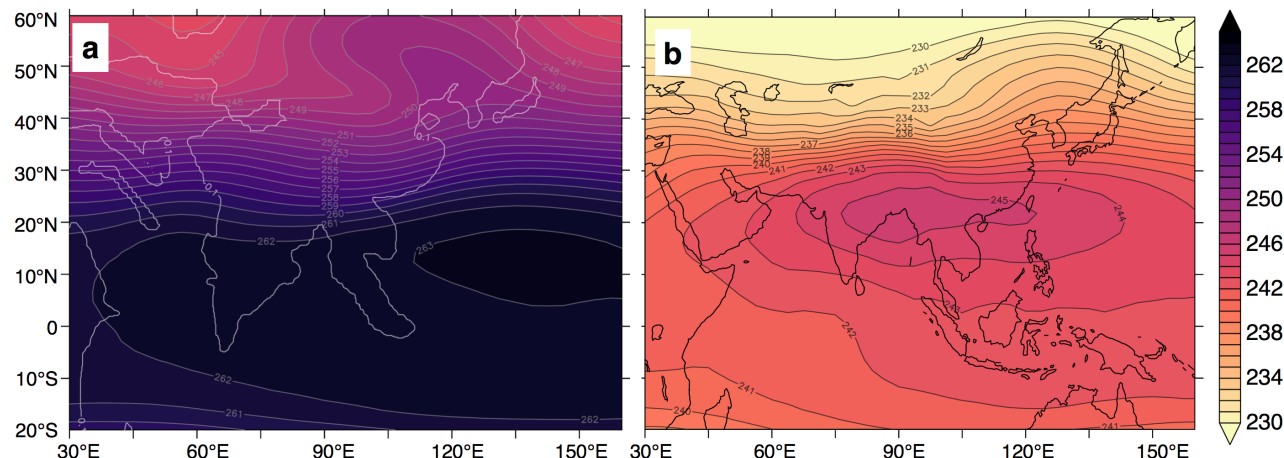

**Figure 9: Application of the Webster and Yang Index (on the region 40 E:110 E, 0:20 N) and comparison of the results obtained for EOC4X (black), control simulation (dotted) and reanalysis (purple).**

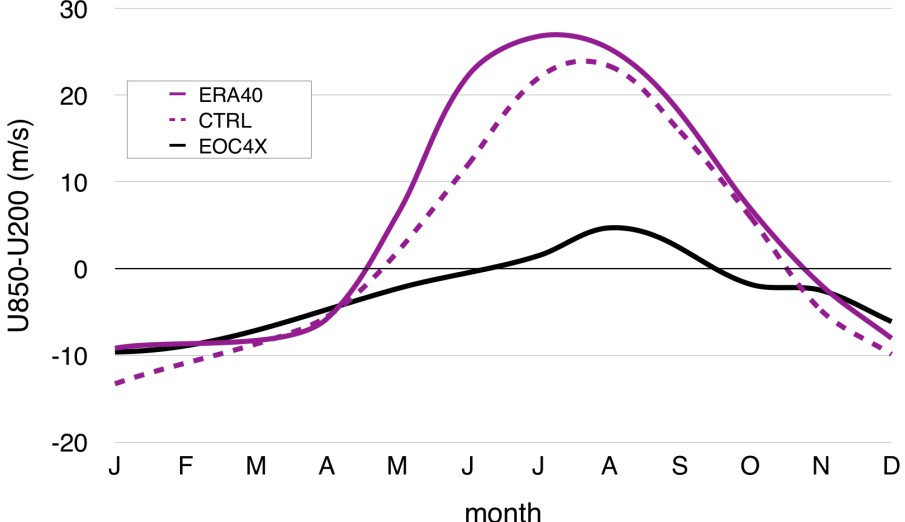

**Figure 10: 3W/3D ratio for EOC4X (a) and Control simulation (b). Regions receiving less than 1mm/day are kept blank. Overlaid magenta outline corresponds to the value 3W3D=5 considered as minimum value in modern monsoonal regions. We also highlight evaporite (red diamonds) and coal deposits (green circles) from Boucot et al. 2013, as well as the five highlighted regions described in the text.**

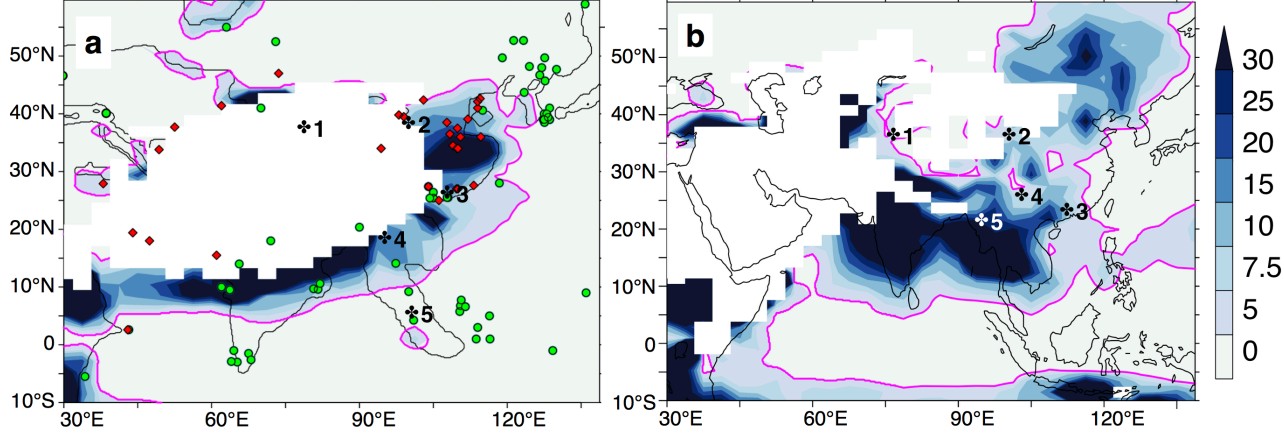

553

## 7. Code availability

LMDZ, XIOS, NEMO and ORCHIDEE are released under the terms of the CeCILL license. OASIS-MCT is released
under the terms of the Lesser GNU General Public License (LGPL). IPSL-CM5A2 code is publicly available through
svn, with the following command lines: svn co
http://forge.ipsl.jussieu.fr/igcmg/svn/modipsl/branches/publications/IPSLCM5A2.1_11192019 modipsl
cd modipsl/util;./model IPSLCM5A2.1

The mod.def file provides information regarding the different revisions used,namely :
– NEMOGCMbranchnemo_v3_6_STABLErevision6665 – XIOS2branchs/xios-2.5revision1763
– IOIPSL/srcsvntags/v2_2_2
– LMDZ5branches/IPSLCM5A2.1rev3591
– branches/publications/ORCHIDEE_IPSLCM5A2.1.r5307rev6336 – OASIS3-MCT2.0_branch(rev4775IPSLserver)

The login/password combination requested at first use to download the ORCHIDEE component is
anonymous/anonymous. We recommend to refer to the project website:
http://forge.ipsl.jussieu.fr/igcmg_doc/wiki/Doc/Config/IPSLCM5A2 for a proper installation and compilation of the
environment.

## 8. Authors contribution

DTB, YD and JBL conducted the Eocene experiments. DTB, FF, YD and GLH analyzed the results and realized the
discussion.
PS, JBL and YD developed the AOGCM version used in this work. FP and GDN reconstructed the Eocene
paleogeography.
The discussion was further emphasized by the contributions of AL (model-data discussion). PS conducted the Control
Simulation and emphasized the model description and the comparison of the Control simulation results with GPCP
observations and ERA40 reanalysis. All co-authors contributed to the writing the manuscript.

## 9. Competing interests

The authors declare that they have no conflict of interest.

## 10. Acknowledgements

The authors thank the two anonymous reviewers and the editor who helped improve the quality of this paper. This work
was granted access to the HPC resources of TGCC under the allocation 2018-A0050107601 made by GENCI. This work

was funded by the INSU-CNRS SYSTER. G. D.-N. acknowledges support from ERC MAGIC grant 649081. This is
IPGP contribution 4115.

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
