# Peer review of "The origin of Asian Monsoons: a modelling perspective"

_Climate of the Past, 2019_

## Referee Comment (RC1) · Anonymous Referee #1 · 16 Jan 2020

In the paper, the authors present their new Late Eocene simulation, and compare the simulation with paleobotanical and sedimentological (coal and evaporites deposits) evidence from Asia. The study is an important contribution for resolving the debate about the timing of Asian monsoon onset, Late Eocene or Late Oligocene/Early Miocene. It demonstrates that 1) the new simulation with IPSL-CM5A2 does not support that modern-like Asian monsoon climate already existed in the Late Eocene, 2) the uncertainties of paleogeography reconstructions (in particular topography and the Tethys reconstructions) have remarkable impacts on the simulated Eocene climate in Asia. This study also fairly points out that the key for finally resolve the debate is collecting the geological data that shows the timing of transition from arid to modern-like monsoonal climate in Asia. The paper is well written. Thus, I suggest accepting the paper

after some minor revisions.

Line 304-305, not only the model bias, but also uncertainties in topography reconstructions, can cause the dry discrepancy in South Asia.

Line 462, the number 5 is missing in the caption.

Line 486, Figure 8 should be replotted. Please check that the purple line does not match with the shaded area in (a). It is better to add the simulated precipitation against with these sedimentological records in the Figure 8, since these records could also reflect dry or wet conditions on the orbital time scale, not only the seasonality.

---

## Referee Comment (RC2) · Anonymous Referee #2 · 8 Feb 2020

Summary:

There is much debate about when modern-like Asian monsoons first formed. This paper investigates the existence of Asian monsoons during the Eocene by an earth system model IPSL-CM5A2. The model simulation shows there were no modern-like Asian monsoons at that time and the authors point out the large uncertainties of paleo-geography result in conflicting modeling results of Asian monsoons in the Eocene. The paper also indicates that most of the current Asian proxy records are located in the seasonal transition zone so they are not ideal for testing the existence of the monsoons. Thus more proxy records and cautious interpretation of the records are necessary to solve the Eocene monsoon puzzle in the future. The paper brings new ideas to the long-debated Eocene monsoons and has sufficient evidence to support it, and it is

well-written. I would like to suggest accepting it after minor revisions.

Comments:

1 The authors conclude there were no modern-like Asian monsoons based on atmospheric circulation (Fig. 4) rather than precipitation seasonality (Fig. 8) in the Eocene simulation. I think it is necessary to have criteria of what atmospheric patterns can be viewed as modern-like monsoons or not. Otherwise, one may argue that Fig. 4c also shows a modern-like monsoon pattern since there is still cross-equator circulation over the Indian ocean though it locates at much lower latitudes.

2 When explaining the Eocene atmospheric circulation (section 3.2), I suggest considering some existing monsoon theories (Boos and Kuang 2010), in which low-level enthalpy or equivalent potential temperature is more physically fundamental to cause circulation and convection anomaly than "blocked by the Tethysian high in Line 267" and "mid-level atmospheric layers very dry and prevents air masses to reach . . ." in Line 280. Generally, we can say that without the blocking of the TP and Iranian Plateau, cold air is easy to intrude the Indian subcontinent and does not allow building up strong positive low-level enthalpy anomaly, thus not triggering much convection as today.

3 I feel like the word "onset" (of Asian monsoons) is confusing. I know that it refers to the beginning of the modern-like monsoons over the geological time scales, but it is also usually used to represent the starting time (day or month) of the summer monsoon season and actually authors use this meaning in Line 144. I suggest replacing "onset" with "origin" or other synonyms.

4 The authors discuss the model-data comparison problem and point out the importance of correct interpretations of paleo-records. One way to better fill in the gaps between model and proxy records is by using isotope-enabled models (e.g., comparing simulated precipitation isotope ratios to proxies based on precipitation isotopes) and proxy forward modeling (e.g., translating climate variables of simulations directly to pseudoproxies). It would be great if authors can add discussion about this.

Line 71: These findings "postpone"... Is it "postpone" or "bring forward"?

Line 75: «doubthouse» -> "doubthouse"

Line 98: "A third mechanism": It is not a mechanism but a conjecture (or other synonyms)

Line 128: expend -> expand

Line 136: improved -> improves

Line 218-221: Cloud feedbacks can also contribute to the model bias: Zhu, J., Poulsen, C. J., & Tierney, J. E. (2019). Simulation of Eocene extreme warmth and high climate sensitivity through cloud feedbacks. Science Advances, 5(9), eaax1874. https://doi.org/10.1126/sciadv.aax1874

Line 270: I don't see easterly winds from the Pacific Ocean

Line 272: Theses-> These

Line 275: How to determine the condensation height? The condensation can occur at multiple layers at a single time in the model.

Line 282: Figure 5->6?

Line 283: "multiple deep convection": how do you identify convection here? By upward motion?

Line 283: add "center" between humidity and around

Line 320-325: Do these records all represent precipitation seasonality/seasonal contrast or annual mean precipitation?

Line 392-393: "When oriented in a NW-SE orientation": change one of the "orient" words

All figures: please enlarge the font size of labels of latitude/longitude/color bar. It is

especially important for Figure 7.

Figure 2: How do you calculate sea level pressure anomaly? Is it seasonal mean minus annual mean? Are winds climatological mean or anomalies?

Line 462: Please add "5" before "Pondaung"

It would be great to add a figure like Figure 5 (a)(b) but in the summer monsoon season in the supplements

―――――――――――――――――――――――

---

## Author Comment (AC1) · 9 Mar 2020

Dear referee,

Thank you for allowing us to go forward in the publication process. We hereby answer to comments and propose a corrected version of the manuscript. You'll find below the answer to the suggested corrections point by point.

Delphine Tardif, on behalf of the co-authors

RC1 : Line 304-305, not only the model bias, but also uncertainties in topography reconstructions, can cause the dry discrepancy in South Asia.

Authors response: We propose the following precision at lines 325-327 : "This could

be attributed either to a bias towards aridity in these specific regions, that is shared by most models (Valdes et al., 2017) and seems to translate in the Eocene as well, and/or to an inaccurate reconstruction of northern Indian late Eocene topography."

RC1 : Line 462, the number 5 is missing in the caption.

Authors response: Done

RC1 : Line 486, Figure 8 should be replotted. Please check that the purple line does not match with the shaded area in (a). It is better to add the simulated precipitation against with these sedimentological records in the Figure 8, since these records could also reflect dry or wet conditions on the orbital time scale, not only the seasonality.

Authors response: Figure 8 is replotted (see below). We also join the proxies against MAP in Supplementary information, Figure 7 (see below).

Figure 10: 3W/3D ratio for EOC4X (a) and Control simulation (b). Regions receiving less than 1mm/day are kept blank. Overlaid purple outline corresponds to the value 3W3D=5 considered as minimum value in modern monsoonal regions. We also highlight evaporite (red diamonds) and coal deposits (green circles) from Boucot et al. 2013, as well as the five highlighted regions described in the text.

[Figure]

**Fig. 1.** Figure 8 replotted (Fig 10 in revised manuscript)

**Figure 7 : Late Eocene Mean Annual Precipitations are shaded (in mm/year) and compared to the occurrence of arid climate related evaporites deposits (red diamonds) and more ever-wet climate related coal deposits (green circles).**

[Figure]

**Fig. 2.** Supplementary Information, Figure 7

---

## Author Comment (AC2) · 9 Mar 2020

Dear referee,

Thank you for allowing us to go forward in the publication process. We hereby answer to comments and propose a corrected version of the manuscript. You'll find below the answers to the suggested corrections and comments point by point.

Delphine Tardif, on behalf of the co-authors

RC2: The authors conclude there were no modern-like Asian monsoons based on atmospheric circulation (Fig. 4) rather than precipitation seasonality (Fig. 8) in the Eocene simulation. I think it is necessary to have criteria of what atmospheric patterns can be viewed as modern-like monsoons or not. Otherwise, one may argue that Fig. 4c

also shows a modern-like monsoon pattern since there is still cross-equator circulation over the Indian ocean though it locates at much lower latitudes.

Authors response: The cross equatorial circulation is indeed simulated at very low latitudes in our late Eocene experiment, over India and SE Asian Peninsula. However, we stress in the paper that this cross-equatorial circulation is deviated to the East at a lower latitude in the Eocene than in the modern climate (Fig. 4c-d). This pattern already dismisses the existence of EAM and suggest a weaker SAM. We also did apply the Webster-Yang Index over the region where the cross-equatorial flow is observed and we showed in the original submitted ms. that values obtained for the late Eocene were significantly lower than modern ones (Lines 355-364 and Figure 9). In order to make our point stronger, we have added the water column integrated moisture flux crossing the equator (Figure 5 in revised ms, see Figure below) where the opposite pattern between the control and the Eocene simulation is clearly visible. In particular, one can see that most moisture transport goes from the Indian ocean to the African continent during the Eocene in the 30°E – 60°E sector. To the contrary, the South East Asian monsoon remains well represented in the Eocene (60-90°E). For full analysis of this new Figure 5, the reviewer is referred to the supplementary document attached to the present response, which shows the modifications made in Section 3.2 (in italic).

RC2: When explaining the Eocene atmospheric circulation (section 3.2), I suggest considering some existing monsoon theories (Boos and Kuang 2010), in which low-level enthalpy or equivalent potential temperature is more physically fundamental to cause circulation and convection anomaly than "blocked by the Tethysian high in Line 267" and "mid-level atmospheric layers very dry and prevents air masses to reach..." in Line 280. Generally, we can say that without the blocking of the TP and Iranian Plateau, cold air is easy to intrude the Indian subcontinent and does not allow building up strong positive low-level enthalpy anomaly, thus not triggering much convection as today.

Authors response: We have added a similar diagnostic as the one proposed by Boos

and Kuang 2010, i.e. the temperature (in °K) at 300 hPa (Figure 8 in the revised version, see Figure below). We show that continental Asia is not the main source of heat for the upper troposphere in the Eocene (a), but rather the western Pacific, which contrasts strongly with the modern case (b, Control Simulation). This also confirms our first interpretations that the High Pressure – Low Pressure zonation and location in the Eocene induces a cascade of events leading to the absence of deep convection over the Himalaya – Tibetan Plateau system. For full analysis of this new Figure 8, the reviewer is referred to the supplementary document attached to this response, which shows the modifications made in Section 3.2 (in italic).

RC2: I feel like the word "onset" (of Asian monsoons) is confusing. I know that it refers to the beginning of the modern-like monsoons over the geological time scales, but it is also usually used to represent the starting time (day or month) of the summer monsoon season and actually authors use this meaning in Line 144. I suggest replacing "onset" with "origin" or other synonyms.

Authors response: We thank the reviewer for this suggestion and have replaced the word "onset" by synonyms

RC2: The authors discuss the model-data comparison problem and point out the importance of correct interpretations of paleo-records. One way to better fill in the gaps between model and proxy records is by using isotope-enabled models (e.g., comparing simulated precipitation isotope ratios to proxies based on precipitation isotopes) and proxy forward modeling (e.g., translating climate variables of simulations directly to pseudoproxies). It would be great if authors can add discussion about this.

Authors response: We thank the reviewer; teh following paragraph has been added in the Conclusion: "Also, rather recent specific modelling techniques could be very promisingly applied as a complement to complex climatic modelling reconstructions. For example, isotopic-enabled models, by simulating paleoprecipitations $\delta 18O$, allow a direct comparison of the model output to $\delta 18O$ values that can be measured in a

wide variety of proxies (shells, carbonates, etc.) and therefore provide robust physical mechanisms to explain the measured patterns (Botsyun et al., 2019; Poulsen et al., 2010). Additionally, the application of proxy forward modelling methods (Dee et al., 2016; Evans et al., 2013), by mimicking the mechanisms through which a particular proxy will record a climatic perturbation (e.g. the translation of a precipitation decrease in an ice core) taking into account the proxy's specificity (e.g. ice compaction and diffusion) and the time uncertainty could contribute greatly to help fill the gap between proxy records and model results."

RC2: Line 71: These findings "postpone"... Is it "postpone" or "bring forward"?

Authors response: They postpone from 22Ma to 40Ma the inception of monsoons

RC2: Line 75: Âńdoubthouse Âż -> "doubthouse"

Authors response: Done

RC2: Line 98: "A third mechanism": It is not a mechanism but a conjecture (or other synonyms)

Authors response: Done, replaced by "theory"

RC2: Line 128: expend -> expand

Authors response: Done

RC2: Line 136: improved -> improves

Authors response: Done

RC2: Line 218-221: Cloud feedbacks can also contribute to the model bias: Zhu, J.,Poulsen, C. J., & Tierney, J. E. (2019). Simulation of Eocene extreme warmth and high climate sensitivity through cloud feedbacks. Science Advances, 5(9), eaax1874.https://doi.org/10.1126/sciadv.aax1874

Authors response: Indeed. We have added this reference and propose the following

correction: "Underlying causes remain unclear and could be attributed to proxy uncertainties, missing processes in the models, (Huber and Caballero, 2011) or biases in the way models handle small-scale processes, such as cloud feedbacks (Zhu et al., 2019)."

RC2: Line 270: I don't see easterly winds from the Pacific Ocean

Authors response: In Fig 4c, the Asian east coast receives westerlies (>30°N) and weak easterlies (<30°C, northern part of Southeast Asian Peninsula). We have clarified the sentence.

RC2: Line 272: Theses-> These

Authors response: Done

RC2: Line 275: How to determine the condensation height? The condensation can occur at multiple layers at a single time in the model.

Authors response: It is the minimal altitude of condensation, corresponding to an approximation of clouds base level

RC2: Line 282: Figure 5->6?

Authors response: Done

RC2: Line 283: "multiple deep convection": how do you identify convection here? By upward motion?

Authors response: Yes

Line 283: add "center" between humidity and around

Authors response: Done

RC2: Line 320-325: Do these records all represent precipitation seasonality/seasonal contrast or annual mean precipitation?
Authors response: They all suggest highly seasonal precipitations. Some also provide Mean Annual Precipitations estimates, but we choose to focus on seasonality because it appears to be a more robust criteria, as explained at Line 393.

RC2: Line 392-393: "When oriented in a NW-SE orientation": change one of the "orient" words

Authors response: Done

RC2: All figures: please enlarge the font size of labels of latitude/longitude/color bar. It is especially important for Figure 7.

Authors response: Done

RC2: Figure 2: How do you calculate sea level pressure anomaly? Is it seasonal mean minus annual mean? Are winds climatological mean or anomalies?

Authors response: Yes and yes, we've modified the Figure legend to be more specific.

RC2: Line 462: Please add "5" before "Pondaung"

Authors response: Done

RC2: It would be great to add a figure like Figure 5 (a)(b) but in the summer monsoon season in the supplements

Authors response: Done, is is now in Figure 5c-d, (see Figure 5 below)

Please also note the supplement to this comment:
https://www.clim-past-discuss.net/cp-2019-144/cp-2019-144-AC2-supplement.pdf

——————————————————

[revised manuscript text omitted]

---

## Author Response (AR2)

Dear Ran,

We thank you for your insightful comments. I detail below the last modifications done to the manuscript considering your remarks. Hope you will find it relevant.

Hope you are safe despite ongoing covid19 outbreak,
Best,

On behalf of the authors,
Delphine Tardif

Editor: Line 44 – 45: "…From this definition, several broad monsoonal regions can be identified over the globe (Zhang and Wang, 2008; Zhisheng et al., 2015), amongst which the Asian Monsoon system, which is itself declined into smaller monsoonal regions (Wang and LinHo, 2002)."
This sentence is a bit awkward.
 Suggested change: …can be identified over the globe (Zhang and Wang, 2008; Zhisheng et al., 2015). A prominent member is the Asian Monsoon System, which covers several smaller monsoonal regions (Wang and LinHo, 2002).
The authors: Thank you for the suggestion, we changed this sentence accordingly (Lines 50-51)

Editor: Line 55 – 56: … it is supposed to be enhanced by orographic insulation…
Please replace "supposed" with "thought".
The authors: Done (Line 61)

Editor: Line 74: The use of "doubthouse" is not that common. Please remove the word "often".
The authors: Done (Line 80)

Editor: Line 133 – 134: please keep the acronyms consistent. "IPSL-CM5A-LR' was not used before.
The authors: Indeed, it was replaced by "IPSL-CM5A", previously introduced at Line 147 (Line 149)

Editor: Line 219 – 220: please also check for studies on changing sea water Uk37 and Mg/Ca values through time, e.g., Tierney & Tingley, (2018, PP) and Tierney et al., (2019 GRL).
The authors: thank you for the suggestion. We propose the following modifications (Lines 237- 242):
« Regarding the proxies, seasonal bias (towards summer or winter, Schouten et al., 2013; Tierney and Tingley, 2018) might affect the temperatures interpreted as representative of the mean annual sea surface temperature. Calibration methods can also be questioned, especially for warmer than present deep-time studies, as it the case for Mg/Ca paleothermometer that doesn't account for the changing Mg/Ca ratio of seawater (Tierney et al., 2019), or for the $U^{K'}_{37}$, for which recent BAYSPLINE recalibration method have proven to lower the mismatch at high temperatures (Tierney and Tingley, 2018). In the case of the $TEX_{86}$, a subsurface bias has been suggested (Ho and Laepple, 2016) and remains debated (Tierney et al., 2017). »

Editor: Line 231 – 232: there were previous studies on the sensitivity of simulated Eocene temperature to western U.S. topography. Two examples are Sewall and Sloan (2006, Geology) and Feng et al., (2016 EPSL).
The authors: thank you, we added the two references (Line 262)

Editor: Line 238: moist adiabatic lapse rate is typically ~5 K/km. 6.5K/km is the global mean lapse rate of present-day. Please clarify.
The authors: 5K/km

Editor: Line 238 – 242: …Considering a moist adiabatic lapse rate of ~6.5°C/km, this suggests the presence of a temperature bias in this region, regardless of the match with modeled values that may themselves be biased…
This sentence is confusing. The conveyed information is a repetition of the previous sentence. Please consider removing it.
The authors: sentence suppressed as suggested below

Editor: Line 330: "...regimes, the amount of precipitations simulated is biased towards aridity, especially…"
Please replace "biased towards aridity" with "underestimated".
The authors: Done (Line 393)

Editor: Line 333: "…hampers a quantitative comparison to paleovegetation records, which mostly provide estimates of required precipitation amounts."
It is unclear what "required precipitation amounts" meant. Please consider removing the whole "which mostly provide estimates of required precipitation amounts.".
The authors: the term "required" may be misleading. We simply explain that paleovegetation studies often provide Mean annual Precipitations values, together with very large uncertainties (that can sometimes double the proposed value). This makes it complicated to compare model precipitation outputs to these Mean Annual Precipitation values retrieved from the data.
We propose to replace "which mostly provide estimates of required precipitations amounts" by "which mostly provide estimates of mean annual precipitations amounts". (Line 396)

Editor: Line 351, Please consider replacing "experiments" with "experiences".
The authors: Done (Line 437)

Editor: Line 445: Please consider replacing "stress out" with "point out".
The authors: Done (Line 537)

Editor: Line 452: Consider replacing "probably intimately" with "closely".
 The authors: Done (Line 544)

Editor: Line 458: Replace "rather recent specific" with "more recently developed".
The authors: Done (Line 554)

Editor: Line 462: …explain the measured patterns (Botsyun et al., 2019; Poulsen et al., 2010). Two additional references are recommended here: Shen et al., (2019, Clim. Past.) and Brady et al., (2019, JAMES)
The authors: thank you for the suggestion, we have added these references as well as Risi et al. 2010, JGR (Line 558)

Editor: Line 462 to 465: The discussion of proxy forward modeling of ice core is out of place. For Deep time, ice core is not an option. Please replace this discussion with something more

deeptime appropriate (e.g., Schmidt 1999, PP; Zhou et al., Zhou et al., 2008 PP; Tindall et al. 2010, EPSL)

The authors: this is correct. However, Zhou et al. 2008 PP and Tindall et al. 2010 EPSL rather describe isotope-enable models for calculation of seawater past d$^{18}$O than integration of a proxy transformation of this signal. Therefore, we suggest to replace the paragraph by the following (Lines 559-564):

"Additionally, the application of proxy forward modelling methods (Evans et al., 2013; Schmidt, 1999), by mimicking the mechanisms through which a particular proxy will record a climatic perturbation (e.g. the translation of water δ$^{18}$O variations by planktonic foraminifera) taking into account the proxy's specificity (e.g. ecology of the foraminifera, episodes of secondary calcification and dissolution) and the time uncertainty could contribute greatly to help fill the gap between proxy records and model results."

Editor: Fig 5: The units for the integrated moisture flux is typically m/s or kg m2/s, an example is Seager and Henderson (2013, J. Clim.). Please check your calculation and write down the equation used here in the figure caption.

The authors: indeed, the unit was wrongly written. The figure displays the northward moisture transport vertically integrated over the whole atmospheric column. It is defined as the mass-weighted vertical integral of the product of northward wind by total water mass per unit mass. The unit was therefore corrected to "kg.m-1.s-1".

Also, we precised in the call for this Figure in the text: "Figure 5 shows the northward moisture transport vertically integrated over the whole atmosphere column for the Control and EOC4X experiments." at Line 305

As well as in the Figure legend: "(a, b) Vertically integrated northward JJA moisture transport averaged between 2°S and 2°N (black lines and left axis);" Line 604

Equation:

$$\vec{Q} = -\frac{1}{g} \int_{p_s}^{0} q\vec{V} dp$$

where q is the specific humidity and V is the horizontal wind vector, g represents gravitational acceleration and ps stands for surface pressure.

Editor: Fig. 8: Please use uniform legend for both subplots given that they are plotting the same variable.

The authors: Done

Editor: Fig. 10 The purple outline shows in magenta color in my screen. Please consider using a different color.

The authors: the color is indeed magenta, it's a confusion in translation, "purple" was replaced by "magenta" in the text. Line 649

Paper: https://doi.org/10.5194/cp-2019-144
Object: Answer to anonymous referee #1

Dear referee,

Thank you for allowing us to go forward in the publication process. We hereby answer to comments and propose a corrected version of the manuscript. You'll find below the answer to the suggested corrections point by point.

Delphine Tardif, on behalf of the co-authors

Line 304-305, not only the model bias, but also uncertainties in topography reconstructions, can cause the dry discrepancy in South Asia.
⟹ We propose the following precision at lines 325-327 : "This could be attributed either to a bias towards aridity in these specific regions, that is shared by most models (Valdes et al., 2017) and seems to translate in the Eocene as well, and/or to an inaccurate reconstruction of northern Indian late Eocene topography."

Line 462, the number 5 is missing in the caption.
⟹ Done

Line 486, Figure 8 should be replotted. Please check that the purple line does not match with the shaded area in (a). It is better to add the simulated precipitation against with these sedimentological records in the Figure 8, since these records could also reflect dry or wet conditions on the orbital time scale, not only the seasonality.
⟹ Figure 8 is replotted. We will join the proxies against MAP in Supplementary information, Figure 7.

Paper: https://doi.org/10.5194/cp-2019-144
Object: Answer to anonymous referee #2

Dear referee,

Thank you for allowing us to go forward in the publication process. We hereby answer to comments and propose a corrected version of the manuscript. You'll find below the answer (in black) to the suggested corrections and comments (in blue) point by point.

Delphine Tardif, on behalf of the co-authors

The authors conclude there were no modern-like Asian monsoons based on atmospheric circulation (Fig. 4) rather than precipitation seasonality (Fig. 8) in the Eocene simulation. I think it is necessary to have criteria of what atmospheric patterns can be viewed as modern-like monsoons or not. Otherwise, one may argue that Fig. 4c also shows a modern-like monsoon pattern since there is still cross-equator circulation over the Indian ocean though it locates at much lower latitudes.

⟹ The cross equatorial circulation is indeed simulated at very low latitudes in our late Eocene experiment, over India and SE Asian Peninsula. However, we stress in the paper that this cross-equatorial circulation is deviated to the East at a lower latitude in the Eocene than in the modern climate (Fig. 4c-d). This pattern already dismisses the existence of EAM and suggest a weaker SAM. We also did apply the Webster-Yang Index over the region where the cross-equatorial flow is observed and we showed in the original submitted ms. that values obtained for the late Eocene were significantly lower than modern ones (Lines 355-364 and Figure 9).

In order to make our point stronger, we have added the water column integrated moisture flux crossing the equator (Lines 290-300, Figure 5) where the opposite pattern between the control and the Eocene simulation is clearly visible. In particular, one can see that most moisture transport goes from the Indian ocean to the African continent during the Eocene in the 30°E – 60°E sector. To the contrary, the South East Asian monsoon remains well represented in the Eocene (60-90°E).

When explaining the Eocene atmospheric circulation (section 3.2), I suggest considering some existing monsoon theories (Boos and Kuang 2010), in which low-level enthalpy or equivalent potential temperature is more physically fundamental to cause circulation and convection anomaly than "blocked by the Tethysian high in Line 267" and "mid-level atmospheric layers very dry and prevents air masses to reach..." in Line 280. Generally, we can say that without the blocking of the TP and Iranian Plateau, cold air is easy to intrude the Indian subcontinent and does not allow building up strong positive low-level enthalpy anomaly, thus not triggering much convection as today.

⟹ We have added a similar diagnostic as the one proposed by Boos and Kuang 2010, i.e. the temperature (in °K) at 300 hPa (see Figure 8 in the revised version). We show that continental Asia is not the main source of heat for the upper troposphere in the Eocene (Fig 8 a), but rather the western Pacific, which contrasts strongly with the modern case (Fig 8b: Control Simulation and SI Fig 6s, ERA5 reanalysis). See Lines 350-354 and Figure 8. This also confirms our first interpretations that the High Pressure – Low Pressure zonation and location in the Eocene induces a cascade of events leading to the absence of deep convection over the Himalaya – Tibetan Plateau system.

I feel like the word "onset" (of Asian monsoons) is confusing. I know that it refers to the beginning of the modern-like monsoons over the geological time scales, but it is also usually used to represent the starting time (day or month) of the summer monsoon season and actually authors use this meaning in Line 144. I suggest replacing "onset" with "origin" or other synonyms.

⟹ We thank the reviewer for this suggestion and have replaced the word "onset" by synonyms

The authors discuss the model-data comparison problem and point out the importance of correct interpretations of paleo-records. One way to better fill in the gaps between model and proxy records is by using isotope-enabled models (e.g., comparing simulated precipitation isotope ratios to proxies based on precipitation isotopes) and proxy forward modeling (e.g., translating climate variables of simulations directly to pseudoproxies). It would be great if authors can add discussion about this.
⟹ We thank the reviewer; a paragraph has been added stating that (l. 516-524):
*"Also, rather recent specific modelling techniques could be very promisingly applied as a complement to complex climatic modelling reconstructions. For example, isotopic-enabled models, by simulating paleoprecipitations $\delta^{18}O$, allow a direct comparison of the model output to $\delta^{18}O$ values that can be measured in a wide variety of proxies (shells, carbonates, etc.) and therefore provide robust physical mechanisms to explain the measured patterns (Botsyun et al., 2019; Poulsen et al., 2010). Additionally, the application of proxy forward modelling methods (Dee et al., 2016; Evans et al., 2013), by mimicking the mechanisms through which a particular proxy will record a climatic perturbation (e.g. the translation of a precipitation decrease in an ice core) taking into account the proxy's specificity (e.g. ice compaction and diffusion) and the time uncertainty could contribute greatly to help fill the gap between proxy records and model results."*

Line 71: These findings "postpone"... Is it "postpone" or "bring forward"?
⟹ They postpone from 22Ma to 40Ma the inception of monsoons

Line 75: «doubthouse» -> "doubthouse"
⟹ Done

Line 98: "A third mechanism": It is not a mechanism but a conjecture (or other synonyms)
⟹ Done

Line 128: expend -> expand
⟹ Done

Line 136: improved -> improves
⟹ Done

Line 218-221: Cloud feedbacks can also contribute to the model bias: Zhu, J.,Poulsen, C. J., & Tierney, J. E. (2019). Simulation of Eocene extreme warmth and high climate sensitivity through cloud feedbacks. Science Advances, 5(9), eaax1874.https://doi.org/10.1126/sciadv.aax1874
⟹ Indeed. We have added this point and this reference (see Line 222-224)

Line 270: I don't see easterly winds from the Pacific Ocean
⟹ In Fig 4c, the Asian east coast receives westerlies (>30°N) and weak easterlies (<30°C, northern part of Southeast Asian Peninsula). We have clarified the sentence (see line 302-303).

Line 272: Theses-> These
⟹ Done

Line 275: How to determine the condensation height? The condensation can occur at multiple layers at a single time in the model.
⟹ It is the minimal altitude of condensation, corresponding to an approximation of clouds base level

Line 282: Figure 5->6?
⟹ Done

Line 283: "multiple deep convection": how do you identify convection here? By upward motion?
⟹ Yes

Line 283: add "center" between humidity and around
⟹ Done

Line 320-325: Do these records all represent precipitation seasonality/seasonal contrast or annual mean precipitation?
⟹ They all suggest highly seasonal precipitations. Some also provide Mean Annual Precipitations estimates, but we choose to focus on seasonality because it appears to be a more robust criteria, as explained at Line 393.

Line 392-393: "When oriented in a NW-SE orientation": change one of the "orient" words
⟹ Done

All figures: please enlarge the font size of labels of latitude/longitude/color bar. It is especially important for Figure 7.
⟹ Done

Figure 2: How do you calculate sea level pressure anomaly? Is it seasonal mean minus annual mean? Are winds climatological mean or anomalies?
⟹ Yes and yes, we've modified the Figure legend to be more specific.

Line 462: Please add "5" before "Pondaung"
⟹ Done

It would be great to add a figure like Figure 5 (a)(b) but in the summer monsoon season in the supplements
⟹ Done, is is now in Figure 5c-d

[revised manuscript text omitted]

